# Beyond Probability Partitions: Calibrating Neural Networks with Semantic Aware Grouping

**Jia-Qi Yang**     **De-Chuan Zhan**\*    **Le Gan**
State Key Laboratory for Novel Software Technology
Nanjing University, Nanjing, 210023, China
yangjq@lamda.nju.edu.cn, zhandc@nju.edu.cn, ganle@nju.edu.cn

## Abstract

Research has shown that deep networks tend to be overly optimistic about their predictions, leading to an underestimation of prediction errors. Due to the limited nature of data, existing studies have proposed various methods based on model prediction probabilities to bin the data and evaluate calibration error. We propose a more generalized definition of calibration error called Partitioned Calibration Error (PCE), revealing that the key difference among these calibration error metrics lies in how the data space is partitioned. We put forth an intuitive proposition that an accurate model should be calibrated across any partition, suggesting that the input space partitioning can extend beyond just the partitioning of prediction probabilities, and include partitions directly related to the input. Through semantic-related partitioning functions, we demonstrate that the relationship between model accuracy and calibration lies in the granularity of the partitioning function. This highlights the importance of partitioning criteria for training a calibrated and accurate model. To validate the aforementioned analysis, we propose a method that involves jointly learning a semantic aware grouping function based on deep model features and logits to partition the data space into subsets. Subsequently, a separate calibration function is learned for each subset. Experimental results demonstrate that our approach achieves significant performance improvements across multiple datasets and network architectures, thus highlighting the importance of the partitioning function for calibration.

## 1 Introduction

With the advancements in deep learning technology, deep learning models have achieved, and in some cases even surpassed, human-level accuracy in various domains[1, 2]. In safety-critical applications[3, 4] such as self-driving[5] and disease diagnosis[6], it is not only desirable to have high accuracy from models but also to have their predicted probabilities reflect the true likelihood of correctness. For instance, if a model predicts a probability of 0.9 for a particular class, we expect the actual probability of being correct to be close to 0.9. This is known as probability calibration. However, recent research has indicated that while deep models have improved in accuracy, their probability calibration has declined[7–9]. Typically, deep models tend to overestimate the probabilities of correct predictions, leading to overconfidence. This discrepancy between accuracy and calibration has made model calibration a crucial research direction.[7, 10–19]

Defining an evaluation method for assessing the degree of calibration is crucial to calibrating a model. Currently, most evaluation methods are based on binning model prediction probabilities, followed by analyzing the distribution of true labels within each bin[7, 12, 20, 21]. The calibration error is then estimated by measuring the difference between the predicted probabilities and the empirical

---

\*De-Chuan Zhan is the corresponding author.

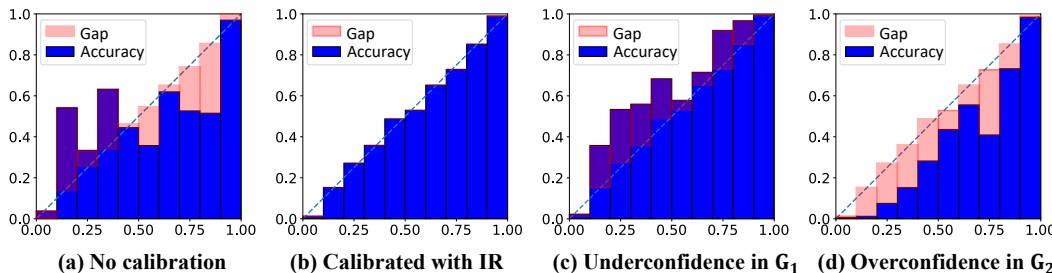

Figure 1: We reconstruct CIFAR10 into a binary classification problem ($[0-4]$ as positives, $[5-9]$ as negatives) and train a Resnet152 on it. Initially, the predicted probabilities are severely uncalibrated, as shown in (a). Then we train an Isotonic regression (IR) model (with the test labels) to calibrate outputs, which leads to nearly perfect calibration in (b). However, the partitions defined by original labels may reveal underlining miscalibration. For example, let $G_1 = \{0, 1, 2, 5, 6, 7\}, G_2 = \{3, 4, 8, 9\}$. The outputs of IR turned out to be significantly underconfident on $G_1$ and overconfident on $G_2$.

distribution. However, such estimation of calibration error fails to reflect the model's accuracy[22]. For instance, consider a binary classification problem where both classes are distributed uniformly, and the model consistently outputs a probability of 0.5 for all inputs. Even using the strictest evaluation method for calibration error, this classifier would appear perfectly calibrated[22]. In reality, as long as the evaluation of calibration error is solely based on prediction probabilities, a classifier with a fixed output marginal distribution $p(y)$ would always be perfectly calibrated but useless. This type of calibration error definition may also be counterintuitive to some extent. For example, a recommendation model may be overconfident in one user group and underconfident in another, yet still be considered calibrated overall. However, no user would perceive the model's output probabilities as accurate in such cases. To illustrate this issue, we present a constructed experiment in Figure. 1, which demonstrates the existence of such situations.

For calibrating a trained model, existing methods mainly rely on transforming the model's predicted probabilities or logits (inputs to the softmax layer)[7, 10–12, 23–25]. For example, histogram binning[7] involves binning the predicted probabilities and calibrating the probabilities within each bin separately. However, as discussed earlier regarding calibration error analysis, if a calibration method relies solely on the model's output probabilities for calibration, it cannot achieve calibration across different semantic contexts ($x$) because the calibration model itself is completely unaware of semantics. Some studies have analyzed calibration across different categories and proposed corresponding calibration methods. For instance, the Dirichlet calibration method[12] models multiclass probabilities as Dirichlet distributions and transforms the uncalibrated probabilities into calibrated ones. These methods incorporate category information, allowing for calibration across different categories. However, this category information may not be directly available, or the known categories may not be the divisions we are interested in. For example, in a recommendation system, we may want the model to be calibrated across various skin color groups, even if we may not have access to the corresponding category information.

**Contribution**. The analysis above has inspired us to propose that a perfectly calibrated model should be calibrated across any data space partition. We present the Partitioned Calibration Error (PCE) as a comprehensive framework for defining semantic-aware calibration errors. By illustrating that various common calibration error metrics are specific instances of PCE under distinct partition functions, we establish a direct correlation between calibration error and model accuracy: PCE converges to accurate score functions of data uncertainty through a bijective grouping function. To discover more effective partition rules, we introduce a technique that incorporates a linear layer onto the deep model's features, enabling the modeling of the partition function. By employing softmax to generate a soft partition, we achieve end-to-end optimization. By generating diverse partition functions, our approach facilitates calibration across different semantic domains, thereby enhancing overall calibration performance without losing accuracy. Experimental results across multiple datasets and network architectures consistently demonstrate the efficacy of our method in improving calibration.

## 2 Partitioned calibration error and grouping-based calibration

We first present our formulation of PCE, which serves as the foundation for our calibration approach.

### 2.1 Measuring calibration with partitions

We start by defining the partition by a grouping function as follows.

**Definition 1** (**Grouping function and input space partition**). *A grouping function $g$ is a mapping from an input $x$ to a group indicator $g(x) = G \in \mathcal{G}$, with the following three properties:*

$$\forall x \in \mathcal{D}, \exists G \in \mathcal{G}, g(x) = G \tag{1}$$

$$\forall x_1, x_2, g(x_1) \neq g(x_2) \rightarrow x_1 \neq x_2 \tag{2}$$

$$\forall \hat{G} \in \mathcal{G}, \exists \hat{x} \in \mathcal{D}, g(\hat{x}) = \hat{G} \tag{3}$$

The group indicator $G$ is also used to denote the induced subset $\{x | g(x) = G\}$.

**Lemma 1.** *A partition of the input space $\mathcal{D}$ is defined by $P = \{\{x | g(x) = \hat{G}\} | \hat{G} \in \mathcal{G}\}$.*

**Proof**. By definition, the subsets in $P$ are (1) non-overlapping (by Eq. (2)), (2) non-empty (by Eq. (3)), (3) the union of all the groups is equal to the universe set(by Eq. (1)). which guarantees $P$ being a valid partition on set $\mathcal{D}$ of all valid $x$. □

We use $x \in G$ and $g(x) = G$ interchangeably, and $(x, y) \in G \leftrightarrow g(x) = G$. With the partition defined above, we can now define the partitioned calibration error.

**Definition 2** (**Partitioned calibration error (PCE)**).

$$\text{PCE}(S, g, \mathcal{L}, f, \mathcal{D}) = \sum_{P \in \mathcal{P}} p(P) \sum_{G \in P} p(G) \mathcal{L}(S(G), S(f(G))) \tag{4}$$

Where $\mathcal{P}$ is the set of all concerned partitions, $p(P)$ is the probability of choosing partition $P$, $p(G) = \int_{(x,y) \in G} p(x, y)$ is the probability of observing a data sample belonging to group $G$, $S(\cdot)$ is a specific statistical magnitude that can be calculated on a group. A straightforward definition of $S(\cdot)$ is the average function $S(G) = \int_{x,y} p_G(x, y) y$, and $S(f(G)) = \int_{x,y} p_G(x, y) f(x)$, where $y$ is the one-hot label with $y_i = 1$ if $x \in i$ th class and $y_i = 0$ otherwise. $f(x)_i$ is the predicted probability of $x$ being $i$th class. $\mathcal{L}(\cdot, \cdot)$ measures the difference between $S(G)$ and $S(f(G))$. $p_G(x, y)$ is the normalized probability density function of $x \in G$, that is,

$$p_G(x, y) = \begin{cases} 0, & \text{if } x \notin G \\ \frac{p(x,y)}{p(G)}, & \text{if } x \in G \end{cases} \tag{5}$$

We will write $p_G(x, y)$ as $p_G$ in the following contents for simplicity. In summary, the aforementioned Eq. (4) defines PCE, which quantifies the expected disparity between the predicted probabilities and the true probabilities within each subgroup, after randomly selecting a partition.

**Example 1** (**Expected calibration error (ECE)**[23]). *With $g(x) = \text{Bin}(f(x))$, where $\text{Bin}(\cdot)$ returns the corresponding Bin ID, $\mathcal{L}(a, b) = |a - b|$, and keep $S$ as the average function.*

$$\text{PCE} = \sum_{G \in P} p(G) \mathcal{L}(S(G), S(f(G))) = \sum_{G \in P} p(G) | \int_{x,y} p_G f(x) - \int_{x,y} p_G y | \tag{6}$$

*and its empirical estimation is*

$$PCE = \sum_G \frac{|G|}{|D|} | \sum_{(x,y) \in G} \frac{1}{|G|} f(x) - \sum_{(x,y) \in G} \frac{1}{|G|} y | \tag{7}$$

*which is exactly the ECE estimator defined in [23] Eq.3.*

Note that in the above ECE, there is only one partition. We provide an example of using multiple partitions in the following example.

**Example 2** (**Class-wise ECE**[12]). *There are $M$ partitions corresponding to $M$ classes. The partition of class $u$ is denoted by $P_u$, the corresponding grouping function is defined by $g_u(x) = \text{Bin}(f(x)_u)$.*

$$\text{Classwise-ECE} = \sum_{P_u} \frac{1}{M} \sum_{G \in P_u} \frac{|G|}{|D|} | \sum_{(x,y) \in G} \frac{1}{|G|} f(x) - \sum_{(x,y) \in G} \frac{1}{|G|} y| \tag{8}$$

*which is exactly the same as Eq. 4 in [12].*

**Example 3** (**Top-label calibration error (TCE)**[25]). *With $g(x) = \text{Bin}(\max_i f(x)_i)$, $S(G) = \frac{1}{|G|} \sum \mathbb{I}(y = \arg\max_i f(x)_i)$, and $S(f(G)) = \frac{1}{|G|} \sum_{(x,y) \sim G} \max_i f(x)_i$. Resulting in the Top-label calibration error defined in [25].*

From the above examples, it can be observed that the sole distinction among the aforementioned calibration metrics lies in the employed grouping function. Hereinafter, we shall delve into two distinctive scenarios of PCE to shed light on the intricate interplay between calibration and accuracy.

**Example 4** (**One-to-one grouping function**). *If the grouping function $g(\cdot)$ is a bijection, then every different $x$ belongs to different groups, which corresponds to point-wise accuracy.*

$$\text{PCE} = \sum_{P \in \mathcal{P}} p(P) \sum_{G \in P} p(G) \mathcal{L}(S(G), S(f(G))) = \int_{x,y} p(x,y) \mathcal{L}(y, f(x)) \tag{9}$$

Minimizing this PCE with bijective grouping function will converge to $f(x) = p(y|x)$ if a proper score function $\mathcal{L}$ (e.g., cross-entropy or Brier score[26]) is used with unlimited data. The uncertainty reflected by $p(y|x)$ is called aleatoric uncertainty[27] and is the best a model can achieve corresponding to Bayes error[28, 29].

**Example 5** (**Constant grouping function**). *If the grouping function $g$ is a constant function with $\forall x_1, x_2, g(x_1) = g(x_2)$, then every different $x$ belongs to a single group.*

$$\text{PCE} = \mathcal{L}\left(\int_{x,y} p(x,y)f(x), \int_{x,y} p(x,y)y\right) \tag{10}$$

*which is minimized by the model outputs the marginal distribution $f(x) = \int_{(x,y)} p(x,y)y = p(y)$.*

We provide proofs and further discussion about Example. (4) and Example. (5) in the supplementary materials. The constant grouping function captures the vanilla uncertainty that we do not have any knowledge about $x$, and we only know the marginal distribution of $y$.

The above analysis demonstrates that by employing finer partitions, PCE becomes a closer approximation to the measure of accuracy. Conversely, coarser partitions align more closely with the traditional calibration metrics utilized in prior research. Since neither extreme partitioning approach applies to practical calibration, selecting an appropriate partitioning method is crucial for calibration performance.

## 2.2 Calibration with sematic-aware partitions

Calibration methods can be designed from the perspective of minimizing the corresponding PCE. For instance, histogram binning[7] adjusts the predicted probabilities within each bin to the mean of the true probabilities in that bin. On the other hand, Bayesian Binning into Quantiles (BBQ)[11] calibration involves setting multiple distinct bin counts and then obtaining a weighted average of the predictions from histogram binning for each bin count, effectively encompassing scenarios with multiple partitions in the PCE. However, existing methods rely on binning and calibration based solely on model output probabilities. Our proposed PCE suggests that the partitioning approach can depend not only on the predicted probabilities but also on the information in input $x$ itself. This would significantly enhance the flexibility of the partitions.

To facilitate implementation, we begin by imposing certain constraints on the range of outputs from the grouping function. We denote the number of partitions by $U = |\mathcal{P}|$ (each partition corresponds to a specific grouping function as defined in Definition 1), and we assume that all data can be partitioned into $K$ groups. The output of the grouping function $g$ takes the form of a one-hot encoding, where if $x$ belongs to the $i$-th group, $g(x)_i = 1, 0 < i < K$, while all other $g(x)_j = 0, j \neq i$. And

$G_i = \{x | g(x)_i = 1\}$ is the $i$-th group. Under these conditions, the expression for PCE can be formulated as follows:

$$\text{PCE} = \sum_i^K \frac{|G_i|}{|D|} \mathcal{L}(S(G_i), S(f_{G_i}(G_i))) \tag{11}$$

From the above equation, it is evident that for each group $G_i$, we can learn a calibration function $f_{G_i}$ specific to that group, which aims to minimize the calibration error within the group $G_i$.

In this paper, we employ the temperature scaling method as the learning approach for the calibration function within each group. Specifically, temperature scaling involves adding a learnable temperature parameter $\tau$ to the logits $o(x)$ (i.e., the input to the softmax function) of the model and applying it to a group, resulting in the following form of grouping loss:

$$\mathcal{L}_{group} = \frac{1}{|G_i|} \sum_{x,y \in G_i} \log \frac{e^{\frac{o(x)_y}{\tau_i}}}{\sum_j e^{\frac{o(x)_j}{\tau_i}}} \tag{12}$$

where the temperature parameter $\tau_i$ is specific to group $G_i$. Note that $\mathcal{L}_{group}$ is not a direct estimation of $\mathcal{L}(S(G_i), S(f_{G_i}(G_i)))$. Specifically, if we choose $S$ to the average function, so the empirical estimation of $S(G_i) = \frac{1}{|G_i|} \sum_{x,y \in G_i} y$, and the empirical estimation of $S(f_{G_i}(G_i)) = \frac{1}{|G_i|} \sum_{x,y \in G_i} f_{G_i}(x)$. Then, we choose the difference measure $\mathcal{L}$ to be the log-likelihood (cross entropy) $\mathcal{L}(S(G_i), S(f_{G_i}(G_i))) = \sum_j S(G_i)_j \log S(f_{G_i}(G_i))_j$, where $j$ is the class index. The $\mathcal{L}_{group}$ will be minimized by $f_{G_i}(x) = y$, which will also minimize $\mathcal{L}(S(G_i), S(f_{G_i}(G_i)))$. Thus, Eq. (12) is a stronger constraint compared with minimizing $\mathcal{L}(S(G_i), S(f_{G_i}(G_i)))$ directly. Our choice of this objective is motivated by two reasons: First, Eq. (12) is able to provide more signals during training since each label $y$ can guide corresponding $f_{G_i}(x)$ directly. On the contrary, if we optimize $\mathcal{L}(S(G_i), S(f_{G_i}(G_i)))$ directly, the labels and predictions are mixed and much label information is lost. Secondly, optimizing Eq. (12) aligns well with the calibration method (TS and ETS) to be used, which leads to better calibration performance.

In the aforementioned equation Eq. (12), we assume that the groups $G_i$ are known. In order to optimize the partition, we incorporate the grouping function $g(\cdot)$ into the loss function. By summing the above equation over all groups according to Eq. (11), we can obtain the final optimization objective.

$$\mathcal{L}_{GC+TS} = \frac{1}{|D|} \sum_{x,y \in D} \log \sum_i g(x)_i \frac{e^{\frac{o(x)_y}{\tau_i}}}{\sum_j e^{\frac{o(x)_j}{\tau_i}}} \tag{13}$$

When the output of $g(x)$ is in the form of one-hot encoding, the above equation represents a standard grouping loss function. To optimize $g$, we introduce a smoothing technique that transforms its output into a probability value. Specifically, we add a linear layer on top of the neural network's features $z$ and follow it with a softmax function as the grouping function $g$, which takes the following form:

$$g_\phi(x) = g'_\phi(z(x)) = \text{softmax}(z(x)\phi_w + \phi_b) \tag{14}$$

where $z \in \mathbb{R}^{|D| \times d_z}$, $\phi_w \in \mathbb{R}^{d_z \times K}$, $\phi_b \in \mathbb{R}^{1 \times K}$. The features $z(x)$ are fixed, which aligns with the post-calibration setting. The above loss function is the combination of grouping and temperature scaling, which is denoted as Grouping Calibration + Temperature Scaling (GC+TS).

**Training $g_\phi(x)$.** The parameters $\phi$ of grouping function $g$ is trained jointly with the calibration function $f_\theta$ on the validation dataset $D_{val}$ (used in early stopping when training deep models). In the GC+TS method, the trainable parameters of $f$ is $\theta = \{\tau_i\}$. In a more intuitive sense, equation Eq. (13) necessitates that the grouping function identifies the optimal partitioning, where adjusting $\tau_i$ within each subgroup $G_i$ minimizes the cross-entropy loss. Furthermore, due to the non-uniqueness of the optimal grouping function that satisfies the aforementioned objective function, we can generate multiple distinct partitions by optimizing with different initial values through multiple iterations. To mitigate the potential overfitting caused by the high flexibility of the grouping function $g$, we have introduced an additional regularization term, and the overall training loss of training $g_\phi$ is $\mathcal{L}_{GC+TS} + \lambda ||\phi_w||_2^2$.

**Training $f_\theta(x)$.** After training of $g_\phi$, we simply apply a calibration method to each group. We can also use temperature scaling in this phase while using hard grouping in Eq. (13). Any other

calibration methods are also applicable, for example, we also adopt Ensemble Temperature Scaling (ETS)[23] method to obtain GC+ETS in our experiments. The calibration function $f_\theta$ is trained on the holdout training dataset $D_{ho}$, while keeping the grouping function fixed. Then the calibration function $f$ is used on each group of the testing dataset. The overall training and predicting procedure is summarized in Algorithm. 1.

**Calibrating with trained partitions**. After training, we partition the input space into different partitions. In each partition, an input $x$ corresponds to a unique group $G$ and a calibration function $f_\theta(x)$, which results in a calibrated probability $p(y|x, G)$. Since there are many different partitions, the groups can be treated as sampled from a distribution $p(G|x)$. We ensemble the calibrated probabilities as the final prediction, which is indeed an Monte-Carlo's estimation of $p(y|x) = \sum_G p(y|x, G)p(G|x)$. This ensemble approach is described in Line 5-11 of Algorithm. 1.

---

**Algorithm 1** Train group calibration with temperature scaling (GC+TS)

---

**Input:** $D_{val} = \{Z_{val}, O_{val}, Y_{val}\}, D_{ho} = \{Z_{ho}, O_{ho}, Y_{ho}\}, D_{test} = \{Z_{test}, O_{test}\}, U, K, \lambda$
**Output:** Calibrated $\hat{Y}_{test}$
 1: **for** $u \leftarrow 0$ to $U - 1$ **do**
 2:     Randomly initialize $\phi_u$ and $\theta_u$
 3:     Optimize $\phi_u$ and $\theta_u$ on $D_{val}$ to minimize Eq. (13) with L-BFGS[30]
 4:     Optimize $\theta_{ui}$ on $D_{ho}$ to minimize Eq. (11) with base calibration method(e.g., TS)
 5:     Calculate partition $P_u = \{G_{ui}\}$ with grouping function $g_{\phi_u}$ on $D_{test}$
 6:     **for** $i \leftarrow 0$ to $K - 1$ **do**
 7:         Calibrate $G_{ui}$ with $f_{\theta_{ui}}(\cdot)$ to obtain $\hat{Y}_{ui}$
 8:     **end for**
 9:     Merge predicts in different groups $\hat{Y}_u = \{\hat{Y}_{ui}\}$
10: **end for**
11: Ensemble predicts in different partitions $\hat{Y}_{test} = \frac{1}{U} \sum_u \hat{Y}_u$

---

**On the usage of $D_{val}$ and $D_{ho}$.** Guo et al. [7] explicitly states that the validation dataset $D_{val}$ used for early stopping can be used for calibration, while most researchers use a hold-out dataset $D_{ho}$ that is not used during training for calibration[12, 23, 25]. Experimentally, we found that calibrating with $D_{val}$ is significantly worse than $D_{ho}$ for existing calibration methods. Interestingly, the grouping function trained on the $D_{val}$ can transfer to $D_{ho}$ for improving performance, which means the training of $g_\phi$ does not require additional data. We investigate this problem in the supplementary in detail.

An essential characteristic of calibration methods is their ability to preserve accuracy[23], ensuring that the classes with the highest probabilities remain unchanged after the calibration process.

**Theorem 1** (**Accuracy-preserving of group calibration**). *Group calibration with any group-wise accuracy-preserving base calibration method is also accuracy-preserving.*

**Proof Sketch.** Since the groups are disjoint, group-wise accuracy-preserving remains overall accuracy-preserving. Ensembles of accuracy-preserving predictions are also accuracy-preserving. $\square$

The Theorem. 1 guarantees our group calibration method remains accuracy-preserving with accuracy-preserving base methods such as temperature scaling[7] or ensemble temperature scaling [23].

## 3 Experiments

To evaluate the performance of our method under various circumstances, we selected three datasets: CIFAR10, CIFAR100[31], and Imagenet[1]. We employed different models for each dataset to reflect our approach's calibration capability across various model accuracy levels[2]. The models used in our experiments include Resnet[2], VGG[32], Densenet[33], SWIN[34], and ShuffleNet[35].

We randomly partitioned a validation set $D_{val}$ from the standard training set: CIFAR10 and CIFAR100 adopted 10% of the data for validation, while Imagenet utilized 5%. Additionally, we randomly sampled 10% of the hold-out data $D_{ho}$ from the standard test set for calibration. We performed 100 different test set splits for each dataset-model combination to obtain reliable results and reported

---

[2]Code and Appendix are available at `https://github.com/ThyrixYang/group_calibration`

the average performance over 100 trials for each method. We conduct paired t-test[36] to measure the static significance of the improvements. The hyperparameters of the comparative methods were tuned based on the corresponding literature with 5-fold cross-validation on the CIFAR10-Resnet152 dataset. We fixed the number of groups at $K = 2$ and the number of partitions at $U = 20$, although 20 is not necessarily the optimal value. The strength of regularization was set to $\lambda = 0.1$, following a similar tuning approach as the comparative methods.

## 3.1 Experimental comparison

Table 1: Comparison of accuracy-preserving calibration methods. We utilized bold font to highlight the statistically superior ($p < 0.01$) results.

| Dataset | Model | Uncal | TS | ETS | IRM(AP) | GC+TS(ours) | GC+ETS(ours) |
|---------|-------|-------|-----|-----|---------|-------------|--------------|
| CIFAR10 | Resnet152 | 0.0249 | 0.0086 | 0.0101 | 0.0115 | **0.0079** | 0.0089 |
| CIFAR10 | Shufflenet | 0.0464 | 0.0107 | 0.0103 | 0.0146 | 0.0099 | **0.0093** |
| CIFAR10 | VGG11 | 0.0473 | 0.0125 | 0.0135 | 0.0157 | **0.0120** | **0.0122** |
| CIFAR100 | Densenet121 | 0.0558 | 0.0418 | 0.0289 | 0.0415 | 0.0411 | **0.0280** |
| CIFAR100 | Resnet50 | 0.0580 | 0.0435 | 0.0292 | 0.0424 | 0.0427 | **0.0269** |
| CIFAR100 | VGG19 | 0.1668 | 0.0485 | 0.0472 | 0.0476 | 0.0414 | **0.0360** |
| Imagenet | Resnet18 | 0.0279 | 0.0178 | 0.0104 | 0.0188 | 0.0173 | **0.0100** |
| Imagenet | Resnet50 | 0.0382 | 0.0182 | **0.0102** | 0.0218 | 0.0174 | **0.0103** |
| Imagenet | Swin | 0.0266 | 0.0367 | 0.0218 | **0.0140** | 0.0366 | 0.0193 |
| Average improvements | | | | | | **-5.03%** | **-8.92%** |

**Accuracy preserving methods**. We compared our approach with methods that offer accuracy assurance, including uncalibrated (Uncal), temperature scaling (TS)[7], Ensemble Temperature Scaling (ETS)[23], and multi-class isotonic regression with accuracy-preserving[3] (IRM(AP))[23] methods. We report the top-label ECE[25]. From Table. (1), it can be observed that our method achieves the best performance on the majority of datasets, and the improvement is statistically significant. Another noteworthy aspect is the enhancement our method demonstrates when compared to the base methods without partitioning, namely GC+TS compared to TS, and GS+ETS compared to ETS. We have provided the average improvement relative to the corresponding base methods in the last row of Table 1. Specifically, GC+TS shows an average improvement of 5.03% over TS, while GC+ETS demonstrates an average improvement of 8.92% over ETS. This indicates that employing better partitioning strategies can enhance calibration performance.

Table 2: Comparison of non-accuracy-preserving calibration methods.

| Dataset | Model | Hist.B | Beta | BBQ | DirODIR | GPC | IRM(NAP) | Acc. Dec. | GC+ETS (ours) |
|---------|-------|--------|------|-----|---------|-----|----------|-----------|---------------|
| CIFAR10 | Resnet152 | 0.0172 | 0.0095 | 0.0097 | 0.0101 | 0.0189 | *0.0087* | -0.079% | **0.0089** |
| CIFAR10 | Shufflenet | 0.0322 | 0.0137 | 0.0139 | 0.0158 | 0.0341 | *0.0119* | -0.050% | **0.0093** |
| CIFAR10 | VGG11 | 0.0279 | 0.0150 | 0.0137 | 0.0156 | 0.0376 | *0.0119* | -0.129% | **0.0122** |
| CIFAR100 | Densenet121 | 0.0632 | 0.0520 | 0.0307 | 0.0533 | 0.0306 | *0.0203* | -0.149% | 0.0280 |
| CIFAR100 | Resnet50 | 0.0618 | 0.0550 | *0.0334* | 0.0553 | 0.0422 | 0.0422 | -3.686% | **0.0269** |
| CIFAR100 | VGG19 | 0.0453 | 0.0642 | *0.0446* | 0.0932 | 0.1406 | 0.0470 | -5.046% | **0.0360** |
| Imagenet | Resnet18 | 0.0714 | 0.0690 | 0.0483 | 0.0386 | -* | *0.0119* | -0.027% | **0.0100** |
| Imagenet | Resnet50 | 0.0502 | 0.0707 | 0.0482 | 0.0326 | - | *0.0107* | -0.006% | **0.0103** |
| Imagenet | Swin | 0.0335 | 0.0629 | 0.0478 | 0.0148 | - | *0.0110* | -0.060% | 0.0193 |

\* GPC failed to converge within a day on Imagenet datasets.

**Non-accuracy preserving methods**. We also compared our method with calibration approaches that do not guarantee accuracy, including Histogram Binning(Hist.B)[7], Beta Calibration (Beta)[10], Bayesian Binning into Quantiles (BBQ)[11], Dirichlet calibration with ODIR regularisation (DirODIR)[12], multi-class isotonic regression without accuracy-preserving (IRM(NAP))[23] and Gaussian process calibration(GPC)[24]. The comparative results are presented in Table. (2). It can be observed that our method achieves improved calibration performance while maintaining the

---

[3]We found that IRM with $\epsilon \ll 10^{-3}$ is indeed not accuracy preserving, so we use $\epsilon = 10^{-3}$ in IRM(AP) and $\epsilon = 10^{-9}$ in IRM(NAP). We discuss this problem in the supplementary material in detail.

same predictive accuracy. On the majority of datasets, our method either meets or surpasses the best non-accuracy-preserving methods. We have also included in Table. (2) the influence of the non-accuracy-preserving method with the lowest ECE on predictive accuracy. It is evident that these methods generally have a noticeable negative impact on model accuracy. In contrast, our proposed method preserves the predictive results, ensuring that the predictive accuracy remains unchanged. We provide all the details and codes of the experiments in the supplementary material, as well as a few additional evaluation metrics (NLL, Birer score) and methods (vector scaling, isotonic regression, etc.)[7], which also supports that our method performs best.

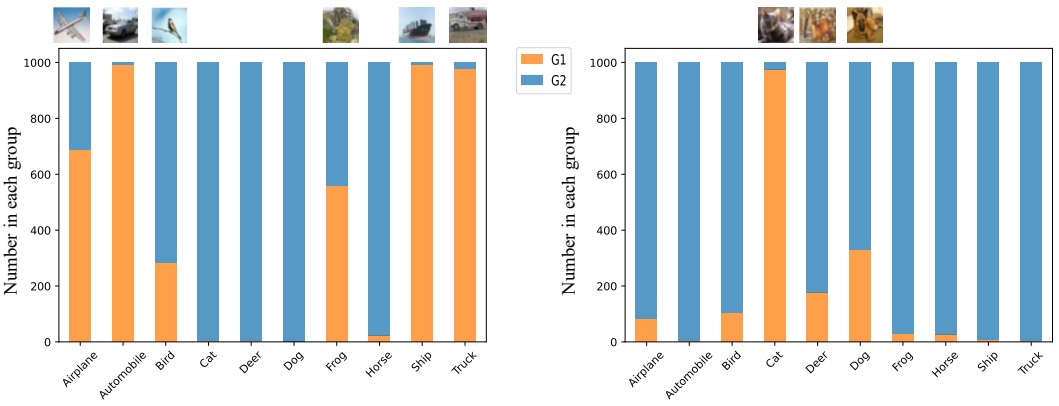

Figure 2: Visualization of learned grouping function on CIFAR10.

## 3.2 Verifying effects of grouping functions

**Quantitative analysis**. We conduct experiments on the constructed setting described in Figure. 1 with TS and our GC+TS method. TS has ECE=0.0157, and GC+TS has ECE=0.0118. We calculate the PCE with the partitions learned by our GC method, and PCE of TS=0.0174, PCE of GC+TS=0.0141, which indicates that GC+TS does improve the overall calibration performance by minimizing PCE.

**Visualization of learned partitions**. To explore the partition functions learned by our method, we visualized the output results of the grouping function on the CIFAR10-Resnet152 dataset. Specifically, we calculated the proportion of each class within each group and visualized two representative partitions in Figure. 2. In the left figure, group 1 primarily consists of airplanes, cars, birds, frogs, ships, and trucks. These classes may share certain visual characteristics like blue sky and water, which implies that the model's predictions might exhibit systematic overconfidence or underconfidence within these categories. Applying a group-specific calibration function to this group can help mitigate miscalibration issues. In the right figure, group 1 mainly comprises cats, deer, and dogs, which also share some visual features. To conclude, our proposed method can implicitly uncover meaningful partitions from the data despite not directly defining a partitioning approach.

## 3.3 Ablation study

We conducted experiments on the CIFAR10-Resnet152 dataset to investigate the impact of three hyperparameters in our method: the number of partitions, the number of groups within each partition, and the regularization strength used in the grouping function. The results are presented in Figure. 3. We observed that as the number of groups decreases, both NLL (negative log-likelihood) and ECE generally exhibit a monotonic increase. We attribute this trend to the fact that increasing the number of groups leads to fewer data points within each group, exacerbating model overfitting. On the other hand, as the number of partitions increases, the calibration performance consistently improves. This finding suggests that training on multiple distinct partitions can enhance the model's performance across various partitions, thereby improving the model's overall performance. This can be viewed as incorporating prior knowledge into the model,

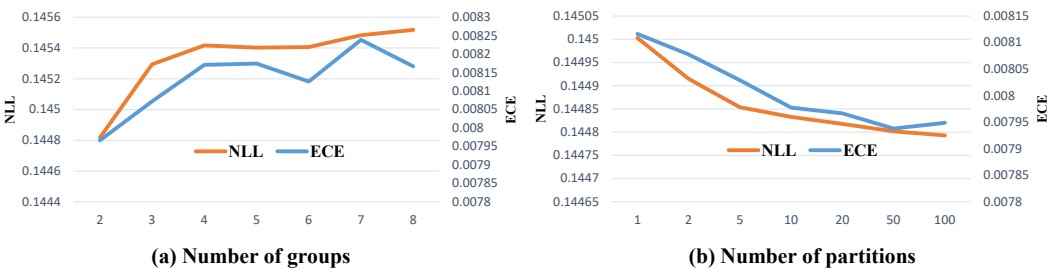

**(a) Number of groups**       **(b) Number of partitions**

Figure 3: The influence of the number of partitions and the number of groups in each partition.

indicating that the model's predictions should be well-calibrated across any partition. The influence of the regularization strength hyperparameter in the grouping function is depicted in Figure. 4. It can be observed that when the regularization strength is too high (i.e., a value of 1), the calibration performance is poorer. This suggests that in such cases, the expressive power of the grouping function is insufficient to learn meaningful groups that aid in calibration. Conversely, when the regularization strength is too low, the performance also deteriorates because the grouping function's expressive power is excessive, leading to the learning of partition functions that are specific to the training set and lack generalization.

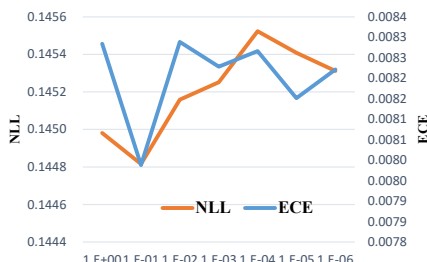

Figure 4: Influence of $\lambda$

## 4 Related Work

We provide a concise overview of recent academic research on the definition, evaluation, and advancements in calibration methods.

**Measures of calibration**. In safety-critical applications[3–6] and tasks involving probability estimation[37–42], it is crucial to ensure that models not only deliver high accuracy but also provide predicted probabilities that accurately represent the genuine likelihood of correctness. As discussed in Section. 2.1, calibration, being a collective concept without direct labels, is typically evaluated by binning the predicted probabilities using different binning approaches. Various studies have proposed different binning methods, each focusing on a different aspect. For example, Kumar et al. [25] proposed binning based on the maximum predicted probability, Kull et al. [12] introduced binning using probabilities for all classes, and Zhang et al. [23], Popordanoska et al. [43], Kumar et al. [44] proposed kernel methods, which can be seen as a smoothed binning approach. In our framework, these methods can be considered as applying different grouping functions, where the grouping functions solely utilize the predicted probabilities. Additionally, other studies have explored alternative characteristics of calibration. For instance, Vaicenavicius et al. [22] and Gupta et al. [14] proposed hypothesis testing methods to assess whether the predicted probabilities approximate calibration, which is still defined solely on the predicted probabilities. Proper scoring rules such as likelihoods and Brier scores are also used to evaluate calibration[26, 45], which are typically used as a training loss to improve calibration.

The calibration considering fairness[46, 47] is also highly related to this work, where the calibration within different populations is considered. Perez-Lebel et al. [48] established an explained component as a metric that lower-bounds the grouping loss in the proper scoring rule theory. However, our work concentrates on learning a better grouping function to improve holistic calibration performance, rather than calibrating some given groups[47].

**Improving calibration of deep models**. The calibration of deep models can be categorized into two main directions. The first direction involves calibrating the model during the training phase by modifying the training process itself[23, 26, 49–51]. This may include using specialized loss functions tailored for calibration[52], employing ensemble methods[26, 53], or data augmentation[54], etc.

However, such approaches require modifications to the training phase and may entail relatively higher computational costs.

The second direction focuses on calibrating the model during the prediction phase. These methods are highly related to the definition of calibration metrics, which involves minimizing the discrepancy between model predictions and aggregated statistics within each bin. The simplest method is histogram binning[7, 37], where predicted probabilities are directly adjusted to match the actual class proportions within each bin. BBQ[11] further proposes an ensemble approach that calibrates results obtained with different numbers of bins. The transformation functions applied to predictions can also be improved beyond averaging within bins. For example, Beta distribution can fit the probability transformation for binary classification[10], while Dirichlet distribution can be employed for multi-class transformation[12]. New families of transformation functions have been proposed to preserve the ordering of predicted probabilities while improving calibration performance[13, 23]. Our method is orthogonal to these calibration methods since the underlying motivation of existing methods is to minimize the calibration error within a single group, while our method offers greater flexibility as we can employ different calibration functions for different groups.

Some recent work also proposed calibration with given groups. Multicalibration[55] proposes an algorithm for learning a multi-calibrated predictor with respect to any given subpopulation class. Durfee et al. [56] proposed to group the feature space with decision trees, then apply Platt scaling within each group, which concentrates on calibration ranking metrics (AUC), while our work concentrates on PCE metrics and proper scoring rules (Log-Likelihood). Yüksekgönül et al. [57] and Xiong et al. [58] propose heuristic grouping functions and apply a specific calibration method within each group, while we aim to propose a learning-based method that can generate partitions in an end-to-end manner.

## 5    Conclusion and limitations

This paper proposes a novel approach to define model uncertainty calibration from the perspective of partitioning the input space, thereby unifying previous binning-based calibration metrics within a single framework. Additionally, we extend the notion of calibration beyond output probabilities, allowing it to be defined on semantically relevant partitions. We establish a connection between calibration and accuracy metrics using semantically relevant partitions. Furthermore, our analysis inspires introducing a method based on learning grouping functions to discover improved partitioning patterns, thus aiding in learning the calibration function. Our framework holds substantial potential for further development, such as optimizing subset selection methods and refining model design, which will be our future work.

**Limitations**. Increasing the number of partitions or employing more complex grouping models will result in higher computational complexity, which is a limitation of our method. Nevertheless, when compared to other Bayesian methods[11, 12, 24], our method demonstrates notably faster speed. We present analysis and experimental comparisons of computational complexities in the supplementary material. Due to the reliance of our method on the features extracted from deep models, it is not applicable for calibrating non-deep models such as tree models[59] and SVMs[12, 60].

## Acknowledgements

This work is supported by the National Key R&D Program of China (2022ZD0114805).

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
