$$= \int_{x,y} p(x, y)\mathcal{L}(y, f(x)) \tag{10}$$

$$= -\int_{x,y} p(x, y) \log f(x)_y \tag{11}$$

$$= -\int_x p(x) \int_y p(y|x) \log f(x)_y \tag{12}$$

$$\geq -\int_x p(x) \left[\int_y p(y|x) \log f(x)_y + \text{KL}(p(y|x), f(x))\right] \tag{13}$$

$$= -\int_x p(x) \left[\int_y p(y|x) \log f(x)_y + \int_y p(y|x) \log p(y|x) - p(y|x) \log f(x)_y\right] \tag{14}$$

$$= -\int_x p(x) \int_y p(y|x) \log p(y|x) \tag{15}$$

Eq. (13) holds because the KL divergence is non-negative $\text{KL}(\cdot, \cdot) \geq 0$, and the optimal loss Eq. (15) is achieved when $\text{KL}(p(y|x), f(x)) = 0$, that is, $p(y|x) = f(x)$. If the Brier score is used, the PCE will become

$$\text{PCE} = \sum_{P \in \mathcal{P}} p(P) \sum_{G \in P} p(G) \mathcal{L}(S(G), S(f(G))) \tag{16}$$

$$= \int_{x,y} p(x,y)(y - f(x))^2 \tag{17}$$

To find the optimal $f(x)$, we take derivative with respect to $f(x)$ and set to 0

$$\frac{\partial PCE}{\partial f(x)} = \int_{x,y} p(x,y) 2(y - f(x)) = 0 \tag{18}$$

$$\int_x p(x) \int_y p(y|x)(y - f(x)) = 0 \tag{19}$$

$$\int_y p(y|x)(y - f(x)) = 0 \tag{20}$$

$$\int_y p(y|x)y = f(x) \int_y p(y|x) \tag{21}$$

$$f(x) = p(y|x) \tag{22}$$

Eq. (20) holds because the derivative should be zero for all the different $x$. In Eq. (21), we use the one-hot encoding of $y$. $\qquad\square$

The uncertainty reflected by $p(y|x)$ is called aleatoric uncertainty[5] and is the best a model can achieve corresponding to Bayes error[6, 7].

**Example 5** (**Constant grouping function**). *If the grouping function $g$ is a constant function with $\forall x_1, x_2, g(x_1) = g(x_2)$, then every different $x$ belongs to a single group.*

$$\text{PCE} = \mathcal{L}\left(\int_{x,y} p(x,y)f(x), \int_{x,y} p(x,y)y\right) \tag{23}$$

*which is minimized by the model outputs the marginal distribution $f(x) = \int_{(x,y)} p(x,y)y = p(y)$.*

*Proof.* If $f(x) = p(y)$, note that $p(y)$ here is a constant vector, so we use $\mathbf{p}_y$ to denote $p(y)$,

$$\int_{x,y} p(x,y)\mathbf{p}_y = \mathbf{p}_y \int_{x,y} p(x,y) \tag{24}$$

$$= \mathbf{p}_y \tag{25}$$

and

$$\int_{x,y} p(x,y)y = \mathbf{p}_y \tag{26}$$

So $f(x) = \mathbf{p}_y$ minimizes Eq. (23). We also need to emphasize that $f(x) = \mathbf{p}_y$ is the simplest minimizer of Eq. (23), but not the unique one. For example, $f(x) = p(y|x)$ also minimizes Eq. (23). $\qquad\square$

The constant grouping function captures the vanilla uncertainty that we do not have any knowledge about $x$, and we only know the marginal distribution of $y$.

An essential characteristic of calibration methods is their ability to preserve accuracy[1], ensuring that the classes with the highest probabilities remain unchanged after the calibration process.

**Theorem 1** (**Accuracy-preserving of group calibration**). *Group calibration with any group-wise accuracy-preserving base calibration method is also accuracy-preserving.*

*Proof.* Since the groups in the same partitions do not overlap, the accuracy-preserving property is trivial with one partition. Assuming model predictions $f(x)$, and we have two partitions with the calibrated predictions $\hat{f}^1(x)$ and $\hat{f}^2(x)$, respectively. If $i$ is the predicted class with the highest probability, that is,

$$\forall j \neq i, f(x)_i > f(x)_j \tag{27}$$

Since we are using accuracy-preserving calibrators within each group, the predicted classes are also $i$ for $\hat{f}^1$ and $\hat{f}^2$,

$$\forall j \neq i, \hat{f}^1(x)_i > \hat{f}^1(x)_j \tag{28}$$

$$\forall j \neq i, \hat{f}^2(x)_i > \hat{f}^2(x)_j \tag{29}$$

Then we take the average of Eq. (28) and Eq. (29)

$$\forall j \neq i, \frac{1}{2}(\hat{f}^1(x)_i + \hat{f}^2(x)_i) > \frac{1}{2}(\hat{f}^1(x)_j + \hat{f}^2(x)_j) \tag{30}$$

which means group calibration with two partitions is accuracy-preserving, the case of more than two partitions can be proved with induction straightforwardly. $\square$

The Theorem. 1 guarantees our group calibration method remains accuracy-preserving with accuracy-preserving base methods such as temperature scaling[8] or ensemble temperature scaling [1].

## 2 On some design choices of group calibration

Due to space constraints, we did not delve into the detailed design aspects of our method in the paper. In this section, we will provide a comprehensive discussion of our rationale behind the selection of the training set for calibration and the choice of group number.

### 2.1 End-to-end training of GC+TS

In our method, we introduced a grouping function, which can be seen as adding complexity to the calibration network. This raises the question of whether the performance improvement of our method is solely due to the increased complexity of the network. This question can be addressed from two aspects.

On the one hand, there have been attempts in the past to use more complex calibration functions, such as the Vector Scaling or Matrix Scaling methods[8]. However, these methods often exhibit poor performance in experiments[8] (also see Table. (8)), primarily due to insufficient data and the risk of overfitting with a large number of parameters. Therefore, simply increasing the complexity of the network does not necessarily improve calibration performance.

On the other hand, we also attempted to directly train the GC+TS method end-to-end, without using the two-stage training approach proposed in the paper, where the grouping function is learned on the validation set and the calibration function is learned on the holdout dataset. The experimental results in Table. (1) showed that this simple end-to-end training approach performed poorly, indicating that it leads to severe overfitting. In contrast, our proposed two-stage training method enables the discovery of better grouping strategies while avoiding the issue of overfitting, thereby improving the overall performance.

### 2.2 The usage of validation dataset

In our comparative experiments, we employed a separate dataset $D_{ho}$ to train the calibration model, which was not used during the training phase. Additionally, our method made use of the validation set $D_{val}$ during training. This raises the question: Does the improved performance of our method stem from the utilization of additional data $D_{val}$? In fact, our experiments revealed that employing the validation set $D_{val}$ in other calibration methods did not consistently enhance the performance on the test set, regardless of whether the validation set was used alone or in conjunction with the hold-out dataset $D_{ho}$. In Table. (2), we have compared the TS and ETS methods using different training sets. It is evident that, in most cases, the ECE trained with a hold-out $D_{ho}$ dataset exhibits lower values compared to the validation set $D_{val}$, particularly noticeable on the Imagenet dataset. Only in a

Table 1: End-to-end training of GC+TS method

| Dataset | Model | GC+TS(e2e) | GC+TS | GC+ETS |
|---------|-------|-----------|-------|--------|
| CIFAR10 | Resnet152 | 0.0105 | **0.0079** | 0.0089 |
| CIFAR10 | Shufflenet | 0.0167 | 0.0099 | **0.0093** |
| CIFAR10 | VGG11 | 0.0169 | **0.0120** | 0.0122 |
| CIFAR100 | Densenet121 | 0.0613 | 0.0411 | **0.0280** |
| CIFAR100 | Resnet50 | 0.0561 | 0.0427 | **0.0269** |
| CIFAR100 | VGG19 | 0.0532 | 0.0414 | **0.0360** |
| Imagenet | Resnet18 | 0.0220 | 0.0173 | **0.0100** |
| Imagenet | Resnet50 | 0.0358 | 0.0174 | **0.0103** |
| Imagenet | Swin | 0.0450 | 0.0366 | **0.0193** |

Table 2: We have listed the performance of TS and ETS trained on the val, ho, and val+ho datasets, along with a comparison to our proposed method. The best performance on each dataset is denoted in italics, while the overall best performance across all methods is indicated in bold.

| Dataset | Model | TS | | | ETS | | | GC+TS(ours) | GC+ETS(ours) |
|---------|-------|-----|-----|--------|-----|-----|--------|-------------|--------------|
| | | val | ho | val+ho | val | ho | val+ho | val+ho | |
| CIFAR10 | Resnet152 | 0.0130 | *0.0086* | 0.0124 | 0.0076 | 0.0101 | ***0.0070*** | 0.0079 | 0.0089 |
| CIFAR10 | Shufflenet | 0.0188 | *0.0107* | 0.0171 | 0.0144 | *0.0103* | 0.0130 | 0.0099 | **0.0093** |
| CIFAR10 | VGG11 | *0.0096* | 0.0125 | 0.0097 | 0.0194 | *0.0135* | 0.0182 | 0.0120 | 0.0122 |
| CIFAR100 | Densenet121 | *0.0383* | 0.0418 | 0.0390 | 0.0274 | 0.0289 | *0.0269* | 0.0411 | 0.0280 |
| CIFAR100 | Resnet50 | 0.0444 | *0.0435* | 0.0442 | *0.0284* | 0.0292 | 0.0290 | 0.0427 | **0.0269** |
| CIFAR100 | VGG19 | *0.0470* | 0.0485 | 0.0481 | 0.0504 | *0.0472* | 0.0496 | 0.0414 | **0.0360** |
| Imagenet | Resnet18 | 0.0192 | *0.0178* | 0.0183 | 0.0174 | *0.0104* | 0.0160 | 0.0173 | **0.0100** |
| Imagenet | Resnet50 | 0.0214 | *0.0182* | 0.0208 | 0.0176 | ***0.0102*** | 0.0165 | 0.0174 | **0.0103** |
| Imagenet | Swin | 0.0461 | *0.0367* | 0.0452 | 0.0222 | *0.0218* | *0.0216* | 0.0366 | **0.0193** |

few instances, employing the val and val+ho sets resulted in improved performance. However, our proposed approach consistently achieves superior performance in the majority of scenarios.

This observation suggests that the performance improvement of our method is not solely attributed to the usage of the validation set $D_{val}$, but rather to our method's ability to identify favorable partitions from the validation set $D_{val}$.

## 2.3 Why not use more than two groups

In our paper, we conducted experiments to investigate the impact of the number of groups, and the results indicated that increasing the number of groups beyond a certain point can lead to a decrease in performance. However, there is a notable improvement when transitioning from one group (TS method) to two groups (GC+TS), which is an intriguing finding. Here, we will qualitatively discuss this phenomenon.

According to our proposed PCE metric, a method should be calibrated across any partition. However, finer-grained partitions can introduce larger estimation errors within each group, and a finer partitioning may not necessarily result in better calibration performance. This is precisely why we proposed learning the grouping function. However, this explanation may not fully elucidate why two groups resulted in the optimal performance, rather than three or four groups. In theory, having more groups should provide the calibration function with greater flexibility, enabling it to learn more effectively.

Indeed, we believe that by fixing the number of groups to two and increasing the number of partitions, we can achieve a balance between the accuracy of statistical estimation and the flexibility of the calibration function. This is because each group within each partition can be different, resulting in each sample having a calibration function belonging to a distinct group with different parameters. Therefore, in practice, the flexibility of the calibration function can increase with the number of partitions. On the other hand, using only two groups can achieve similar effects to having more than two groups. For example, having a single partition with groups [[1], [2], [3]] is similar to having three partitions with groups [[1], [2, 3]], [[2], [1, 3]], and [[3], [1, 2]], respectively. In both cases, they constrain the calibration errors within the groups [1], [2], and [3].

In summary, we believe that increasing the number of partitions is more effective than increasing the number of groups in achieving better results.

# 3 Measuring the accuracy preserving performance

In existing research, it is generally believed that preserving accuracy entails ensuring that the calibrated probabilities of the most probable class remain consistent with the pre-calibrated probabilities[1]. However, we have discovered certain limitations to this criterion in practical applications. Specifically, when the calibrated probabilities of the highest and second-highest classes are extremely close, it becomes challenging to determine if the difference between them can be utilized for the final decision-making process. While numerically we can still identify the class with the highest probability, intuitively, this prediction outcome has already lost the ability to single out the class with the highest probability and instead yields multiple classes with equally high possibilities.

We have observed that all accuracy-preserving calibration methods exhibit a reduction in the gap between the highest and second-highest probabilities after calibration, which aligns with our expectations. This is because we believe deep models generally tend to be overly optimistic, necessitating a decrease in the probabilities assigned to highly predicted classes. Table. (3) reports the differences in predicted probabilities (highest - second highest) for the TS, ETS, and IRM methods. It can be observed that all three methods reduce the optimism in predictions. Specifically, we report the predicted probability differences for the IRM method with epsilon values of 1e-3, 1e-4, and 1e-9, with 1e-9 being the recommended value in the article. We found that as the epsilon value decreases, the accuracy-preserving capability of the IRM method gradually weakens, resulting in smaller differences. This, in turn, leads to an increasing indistinguishability among certain prediction outcomes.

Table 3: The average difference between the largest and second-largest predicted probabilities.

| Dataset | Model | Uncal | TS | ETS | IRM(9) | IRM(4) | IRM(3) | GC+TS | GC+ETS |
|---|---|---|---|---|---|---|---|---|---|
| CIFAR10 | Resnet152 | 0.966660 | 0.939089 | 0.930004 | 0.938682 | 0.938685 | 0.938710 | 0.938935 | 0.930816 |
| CIFAR10 | Shufflenet | 0.924464 | 0.866725 | 0.857069 | 0.867618 | 0.867623 | 0.867675 | 0.866727 | 0.857091 |
| CIFAR10 | VGG11 | 0.945276 | 0.895988 | 0.875528 | 0.888638 | 0.888644 | 0.888694 | 0.895414 | 0.877650 |
| CIFAR100 | Densenet121 | 0.781146 | 0.757266 | 0.734962 | 0.745235 | 0.745239 | 0.745272 | 0.757469 | 0.734396 |
| CIFAR100 | Resnet50 | 0.789668 | 0.765334 | 0.737859 | 0.749342 | 0.749346 | 0.749383 | 0.765930 | 0.737448 |
| CIFAR100 | VGG19 | 0.851717 | 0.712838 | 0.675551 | 0.715064 | 0.715079 | 0.715209 | 0.713981 | 0.676106 |
| Imagenet | Resnet18 | 0.607665 | 0.579059 | 0.579886 | 0.589781 | 0.589782 | 0.589798 | 0.579222 | 0.580189 |
| Imagenet | Resnet50 | 0.690282 | 0.652209 | 0.650268 | 0.664278 | 0.664281 | 0.664304 | 0.652457 | 0.650676 |
| Imagenet | Swin | 0.703577 | 0.764912 | 0.726672 | 0.722184 | 0.722182 | 0.722164 | 0.765201 | 0.727472 |

To explore the impact of this effect on prediction outcomes, we report the minimum value of (highest probability - second highest probability) on the test set in Table. (4). It can be observed that the TS, ETS, GC+TS, and GC+ETS methods maintain a similar magnitude for this difference as compared to the uncalibrated values, while IRM significantly reduces this difference in magnitude. When the epsilon value is set to 1e-9, the difference between the two probabilities becomes indistinguishable and evaluates to zero using 32-bit floating-point representation. This phenomenon is more pronounced in the Imagenet dataset, where even with an epsilon value of 1e-4, differentiation is not possible. Although an epsilon value of 1e-3 ensures differentiation in floating-point precision, the difference is reduced to the order of 1e-8. In light of this observation, we classify IRM with eps=1e-3 as accuracy-preserving and IRM with eps=1e-9 as non-accuracy-preserving in our paper. It is important to emphasize that even with eps=1e-3, IRM significantly weakens the discriminative ability of prediction probabilities, and its accuracy-preserving capability is inferior to the other methods mentioned.

# 4 Training complexity

Due to significant differences in implementation among the various methods, directly analyzing complexity may not accurately reflect their actual running speeds. For instance, methods involving extensive matrix operations, such as matrix scaling, may have substantial computational requirements, but they can leverage GPU parallelization for efficient execution. On the other hand, methods like histogram binning may have lower computational requirements, but their calculations are difficult to parallelize, resulting in potentially longer actual execution times.

Table 4: The minimum difference between the largest and second-largest predicted probabilities.

| Dataset | Model | Uncal | TS | ETS | IRM(9) | IRM(4) | IRM(3) | GC+TS | GC+ETS |
|---|---|---|---|---|---|---|---|---|---|
| CIFAR10 | Resnet152 | 1.795E-03 | 1.073E-03 | 1.004E-03 | 0.000E+00 | 1.909E-07 | 1.985E-06 | 1.066E-03 | 9.897E-04 |
| CIFAR10 | Shufflenet | 2.641E-04 | 1.534E-04 | 1.458E-04 | 0.000E+00 | 2.563E-08 | 3.016E-07 | 1.611E-04 | 1.590E-04 |
| CIFAR10 | VGG11 | 1.954E-03 | 1.122E-03 | 9.833E-04 | 0.000E+00 | 1.802E-07 | 1.783E-06 | 1.015E-03 | 9.055E-04 |
| CIFAR100 | Densenet121 | 1.621E-04 | 1.304E-04 | 1.282E-04 | 0.000E+00 | 8.345E-09 | 1.211E-07 | 1.414E-04 | 1.317E-04 |
| CIFAR100 | Resnet50 | 5.580E-05 | 4.733E-05 | 4.515E-05 | 0.000E+00 | 2.980E-10 | 5.588E-08 | 4.985E-05 | 4.646E-05 |
| CIFAR100 | VGG19 | 3.596E-04 | 1.862E-04 | 1.633E-04 | 0.000E+00 | 2.667E-08 | 2.839E-07 | 1.617E-04 | 1.387E-04 |
| Imagenet | Resnet18 | 2.672E-06 | 2.142E-06 | 2.295E-06 | 0.000E+00 | 0.000E+00 | 3.725E-10 | 2.178E-06 | 2.334E-06 |
| Imagenet | Resnet50 | 8.310E-06 | 5.745E-06 | 6.232E-06 | 0.000E+00 | 0.000E+00 | 1.341E-09 | 5.946E-06 | 6.426E-06 |
| Imagenet | Swin | 8.106E-05 | 1.022E-04 | 8.781E-05 | 0.000E+00 | 5.215E-10 | 6.765E-08 | 8.153E-05 | 8.153E-05 |

To address this, we conducted experiments to compare the running speeds of different methods. In our comparisons, we utilized the open-source implementations provided in the original papers of each method to ensure that the algorithm implementations themselves were adequately optimized. To ensure fairness in the comparison, we employed the same hardware across all experiments (Nvidia RTX 2080Ti GPU, Intel(R) Xeon(R) Silver 4110 CPU, 120 GB memory).

Figure. 1 presents a comparison of the training times for various calibration methods on the CIFAR10 and ImageNet datasets. We can observe that on the CIFAR10 dataset, most of the CPU-based methods such as histogram binning and beta calibration have very short running times (less than a second) and can be considered negligible. On the other hand, methods that require GPU computations, such as GC+TS and matrix scaling, generally exhibit slower speeds but still complete in less than a minute. This is partly due to the time required for GPU communication in GPU-based methods, and also because the parallelization advantages of these methods become more pronounced with larger datasets.

On the ImageNet dataset, the speed differences among various methods become more pronounced. This can be attributed to the larger number of data points and the increased number of classes in ImageNet. Among all the compared methods, IRM and ETS exhibit the fastest speeds, while DirODIR and BBQ methods are the slowest. The slowest method (DirODIR) takes approximately 50 minutes, which is still acceptable considering the size of the ImageNet dataset. In contrast, our methods, GC+TS and GC+ETS, require only approximately 100 seconds and 300 seconds, respectively. This demonstrates their capability to handle datasets several orders of magnitude larger or accommodate increased numbers of partitions and groups. Therefore, computational complexity is not a bottleneck for our methods.

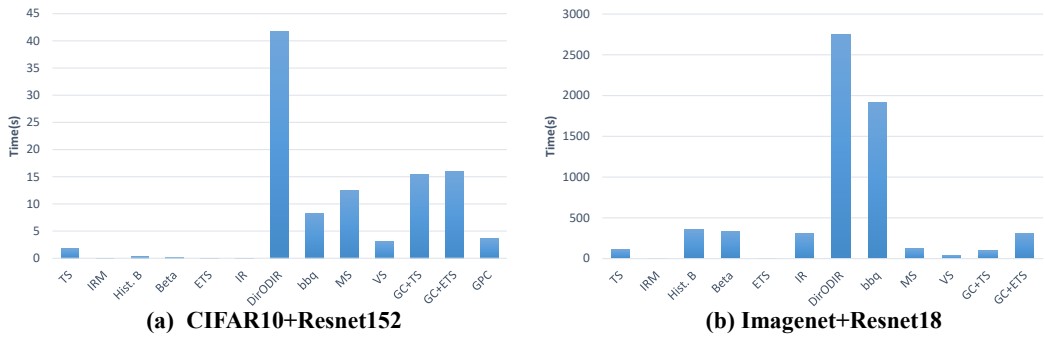

(a) CIFAR10+Resnet152        (b) Imagenet+Resnet18

Figure 1: Comparison of the training times for various calibration methods on CIFAR10 and Imagenet datasets.

# 5 Other metrics and analysis

Due to the limitations of ECE in capturing accuracy, existing research also recommends the concurrent use of proper scoring rules to measure the effectiveness of calibration. In the following analysis, we compare the negative log-likelihood and Brier score of each comparative method.

In Table. (5) and Table. (6), we observe that our methods, GC+TS and GC+ETS, achieve the best performance on the majority of datasets. This finding provides further evidence that our approach attains superior calibration performance on the whole by optimizing calibration metrics across different partitions. It is worth noting that we observed a weak discriminative power of the Brier Score among the various methods. The performance of most methods was quite similar in terms of the Brier Score. Therefore, we believe that although both the Negative Log-Likelihood (NLL) and Brier Score are proper scoring rules, the NLL is better suited for measuring calibration performance.

Table 5: Negative log-likelihood (NLL) of accuracy-preserving methods.

| Dataset | Model | Uncal | TS | ETS | IRM(AP) | GC+TS(ours) | GC+ETS(ours) |
|---|---|---|---|---|---|---|---|
| CIFAR10 | Resnet152 | 0.1747 | 0.1461 | 0.1489 | 0.1569 | **0.1448** | 0.1482 |
| CIFAR10 | Shufflenet | 0.3164 | 0.2648 | 0.2666 | 0.2782 | **0.2643** | 0.2670 |
| CIFAR10 | VGG11 | 0.3133 | 0.2576 | 0.2616 | 0.2656 | **0.2555** | 0.2595 |
| CIFAR100 | Densenet121 | 0.8335 | 0.8272 | 0.8463 | 0.8299 | **0.8251** | 0.8437 |
| CIFAR100 | Resnet50 | 0.8348 | 0.8284 | 0.8484 | **0.8205** | 0.8236 | 0.8431 |
| CIFAR100 | VGG19 | 1.3939 | 1.1241 | 1.1303 | 1.1237 | **1.1151** | 1.1222 |
| Imagenet | Resnet18 | 1.2729 | 1.2659 | 1.2757 | 1.2688 | **1.2648** | 1.2752 |
| Imagenet | Resnet50 | 1.0126 | 0.9988 | 1.0094 | 1.0020 | **0.9972** | 1.0086 |
| Imagenet | Swin | 0.8255 | 0.7981 | 0.8102 | **0.7550** | 0.7963 | 0.8087 |

Table 6: Brier score of accuracy-preserving methods.

| Dataset | Model | Uncal | TS | ETS | IRM(AP) | GC+TS(ours) | GC+ETS(ours) |
|---|---|---|---|---|---|---|---|
| CIFAR10 | Resnet152 | 0.0069739 | 0.0066344 | 0.0066347 | 0.0067082 | **0.0066010** | 0.0066075 |
| CIFAR10 | Shufflenet | 0.0137624 | 0.0129163 | 0.0129165 | 0.0130136 | **0.0128871** | 0.0128920 |
| CIFAR10 | VGG11 | 0.0126271 | 0.0117278 | 0.0116770 | 0.0117384 | 0.0116741 | **0.0116410** |
| CIFAR100 | Densenet121 | 0.0030042 | 0.0029858 | 0.0029806 | **0.0029726** | 0.0029834 | 0.0029778 |
| CIFAR100 | Resnet50 | 0.0030039 | 0.0029786 | 0.0029702 | **0.0029536** | 0.0029704 | 0.0029628 |
| CIFAR100 | VGG19 | 0.0042033 | 0.0037378 | 0.0037180 | 0.0037122 | 0.0037116 | **0.0036893** |
| Imagenet | Resnet18 | 0.0004200 | 0.0004199 | 0.0004195 | 0.0004200 | 0.0004197 | **0.0004193** |
| Imagenet | Resnet50 | 0.0003499 | 0.0003489 | 0.0003485 | 0.0003490 | 0.0003486 | **0.0003483** |
| Imagenet | Swin | 0.0002833 | 0.0002847 | 0.0002826 | **0.0002801** | 0.0002842 | 0.0002821 |

In Table. (7), we also compared the methods based on the Maximum Calibration Error (MCE)[8], and we observed that our method performed well on most datasets. The MCE demonstrated a higher level of discrimination, showing significant differences among the methods. However, we believe that the MCE may not align well with most real-world applications as it only reflects the maximum error in predicted probabilities. Therefore, its relevance is limited to specific application scenarios and may not provide comprehensive insights into overall calibration performance.

Table 7: MCE of accuracy-preserving methods.

| Dataset | Model | Uncal | TS | ETS | IRM(AP) | GC+TS(ours) | GC+ETS(ours) |
|---|---|---|---|---|---|---|---|
| CIFAR10 | Resnet152 | 0.3174 | 0.2475 | **0.2249** | 0.2336 | 0.2469 | 0.2294 |
| CIFAR10 | Shufflenet | 0.7058 | 0.1641 | 0.3548 | 0.2906 | **0.1469** | 0.1779 |
| CIFAR10 | VGG11 | 0.2693 | **0.1372** | 0.2676 | 0.1930 | 0.1580 | 0.2203 |
| CIFAR100 | Densenet121 | 0.1402 | 0.1189 | 0.1608 | 0.1451 | **0.1175** | 0.1527 |
| CIFAR100 | Resnet50 | 0.1846 | 0.2370 | 0.2811 | 0.4409 | **0.1621** | 0.2280 |
| CIFAR100 | VGG19 | 0.4546 | 0.1799 | 0.1267 | 0.1503 | 0.1881 | **0.1185** |
| Imagenet | Resnet18 | 0.0511 | 0.0535 | 0.0473 | **0.0432** | 0.0575 | 0.0486 |
| Imagenet | Resnet50 | 0.0708 | 0.0559 | 0.0487 | 0.0498 | 0.0541 | **0.0467** |
| Imagenet | Swin | 0.0478 | 0.1149 | 0.0476 | 0.0479 | 0.1008 | **0.0454** |

In Table. (8), we reported the Negative Log-Likelihood (NLL) of the non-accuracy-preserving methods. Since NLL is a proper scoring rule, it can reflect both calibration and accuracy. It can be observed that our method exhibits a clear advantage in terms of NLL, indicating that our approach can effectively balance calibration and accuracy.

Table 8: Negative log-likelihood of non-accuracy-preserving methods.

| Dataset | Model | Hist.B | Beta | BBQ | DirODIR | GPC | IRM(NAP) | GC+TS(ours) |
|---------|-------|--------|------|-----|---------|-----|----------|-------------|
| CIFAR10 | Resnet152 | 0.8395 | 0.1633 | 0.2251 | 0.1554 | 0.1581 | 0.1626 | **0.1448** |
| CIFAR10 | Shufflenet | 1.2102 | 0.2730 | 0.3757 | 0.2797 | 0.2882 | 0.2828 | **0.2643** |
| CIFAR10 | VGG11 | 1.1258 | 0.2656 | 0.4001 | 0.2658 | 0.2816 | 0.2700 | **0.2555** |
| CIFAR100 | Densenet121 | 6.0120 | 1.2249 | 1.5252 | 0.8969 | 0.8129 | 0.8358 | **0.8251** |
| CIFAR100 | Resnet50 | 5.7539 | 1.2165 | 1.4939 | 0.8957 | 0.8112 | 0.8257 | **0.8236** |
| CIFAR100 | VGG19 | 4.6565 | 1.5075 | 2.1336 | 1.3645 | 1.2768 | 1.1249 | **1.1151** |
| Imagenet | Resnet18 | 11.2355 | 2.2905 | 2.2860 | 1.6486 | - | 1.2689 | **1.2648** |
| Imagenet | Resnet50 | 9.6245 | 2.0287 | 1.9661 | 1.3335 | - | 1.0020 | **0.9972** |
| Imagenet | Swin | 9.8436 | 1.7640 | 1.8490 | 0.8946 | - | **0.7558** | 0.7963 |

Due to space limitations, we did not include the results of some less effective comparative methods in the paper. In Table. (9), we reported the results of the Vector Scaling, Matrix Scaling, and Isotonic Regression methods[8]. These methods are not accuracy-preserving, and their ECE values are significantly worse compared to our proposed method.

Table 9: ECE of vector scaling, matrix scaling and isotonic regression compared with our method.

| Dataset | Model | Vec. S | Mat. S | IR | GC+TS(ours) | GC+ETS(ours) |
|---------|-------|--------|--------|-----|-------------|--------------|
| CIFAR10 | Resnet152 | 0.0088 | 0.0133 | 0.0137 | **0.0079** | 0.0089 |
| CIFAR10 | Shufflenet | 0.0118 | 0.0180 | 0.0267 | 0.0099 | **0.0093** |
| CIFAR10 | VGG11 | 0.0133 | 0.0187 | 0.0205 | **0.0120** | 0.0122 |
| CIFAR100 | Densenet121 | 0.0578 | 0.2436 | 0.0772 | 0.0411 | **0.0280** |
| CIFAR100 | Resnet50 | 0.0622 | 0.2373 | 0.0862 | 0.0427 | **0.0269** |
| CIFAR100 | VGG19 | 0.0538 | 0.2636 | 0.1307 | 0.0414 | **0.0360** |
| Imagenet | Resnet18 | 0.0477 | 0.5002 | 0.1701 | 0.0173 | **0.0100** |
| Imagenet | Resnet50 | 0.0528 | 0.4289 | 0.1532 | 0.0174 | **0.0103** |
| Imagenet | Swin | 0.0708 | 0.1876 | 0.1900 | 0.0366 | **0.0193** |

# 6 Reproducibility

To ensure the reproducibility of our paper, we have included all the necessary code for replicating the experiments in the supplementary materials. The code has been anonymized to maintain the anonymity of the review process. Instructions for running the code and specific implementation details for each method can be found in the README.md file and commented within the code itself.

**Implementation and hyper-parameter tuning**. The implementations of the comparative methods in the paper have been modified from the corresponding open-source codes of their respective papers. Specifically, the DirODIR[2] and Vector Scaling methods were adapted from the open-source code[2] of Kull et al. [2]. The BBQ and GPC methods were modified from the open-source code[3] of Wenger et al. [9]. The ETS and IRM methods were adapted from the open-source code[4] of Zhang et al. [1]. The Temperature Scaling, Histogram Binning, Beta Calibration, and Isotonic Regression methods were modified from the open-source code[5] of Guo et al. [8]. For some hyperparameters in the comparative methods, we conducted parameter tuning on the CIFAR-10 dataset. It is worth noting that the DirODIR[2] method employs cross-validation to adjust the parameters mu and lambda on all datasets, with over 100 combinations of parameter values. This search process incurs significant time overhead. Considering that the DirODIR method itself is slower than the other methods, we believe

---

[2] `https://github.com/dirichletcal/experiments_dnn/blob/master/scripts/calibration/cal_methods.py`

[3] `https://github.com/JonathanWenger/pycalib/tree/master/pycalib`

[4] `https://github.com/zhang64-llnl/Mix-n-Match-Calibration/blob/master/util_calibration.py`

[5] `https://github.com/markus93/NN_calibration/blob/master/scripts/calibration/cal_methods.py`

that conducting hyperparameter search for each dataset is not equitable. Therefore, for the DirODIR method, we follow the same approach as the other methods by tuning the parameters on one dataset and applying the same set of parameters to the other datasets. This is one of the reasons why the performance of the DirODIR method is notably lower than the results reported in the original paper.

**Datasets**. Due to the size limitations of the supplementary materials, we have only uploaded the CIFAR10-Resnet152 dataset. Upon acceptance of the paper, we will make all the code and the remaining datasets publicly available together. We also provide all the code for training the deep networks and extracting features.