# OpenReview forum: "Beyond probability partitions: Calibrating neural networks with semantic aware grouping"
_NeurIPS.cc/2023/Conference — NeurIPS 2023 poster_

### Official Review · Reviewer_mMe6 · 2023-06-21

**Soundness:** 1 poor
**Presentation:** 2 fair
**Contribution:** 2 fair
**Rating:** 5
**Confidence:** 4

**Summary:**

The paper proposes a novel method for dealing with the well-known problem of miscalibration in deep learning models (and machine learning models more generally). The proposed approach is to partition the input space and fit a calibration function to each set. The paper shows that the idea of partitioning the input space generalizes two special cases --- one-to-one grouping and constant grouping. Experiments are conducted for image classification on the CIFAR10, CIFAR100 and ImageNet datasets.

**Strengths:**

I really like what the paper is trying to do. Unfortunately the mathematical presentation contains many technical errors and I have concerns about the theoretical underpinning of the proposed approach as discussed below.

**Weaknesses:**

I have two main concerns about the paper. First, the mathematical presentation contains many technical errors:

1. For Lemma 1 to be correct we need the additional assumption that
$\forall x \in {\cal D}, \exists G \in {\cal G} \text{ s.t. } g(x) = G.$
This should be explicitly stated.

2. In Definition 2 (Lines 84--90) there appears to be inconsistencies in the definition of a group G. On Line 84 we have $(x, y) \in G$, on Line 90 we have $x \in G$, and on Line 78 we have $g(x) = G$. I understand that the authors may be overloading the notation here so that $x \in G$ implies $g(x) = G$, but it remains to define what $(x, y) \in G$ means. The paper should be rigorous in its mathematical definitions and notation.

3. Equation 13, $g_\phi(x) = g_\phi(z(x))$ makes no sense unless $z$ is the identity mapping (or $g_\phi$ is constant). Moreover, $z$ is defined to be a matrix and given its dimensionality, $|D| \times d_z$, it is unclear what $z(x)$ actually does.

I am also concerned about the correctness of the theoretical idea that underpins the approach. Specifically, we can think of the neural network classifier as computing $p(y \mid x)$, which we wish to be calibrated. By partitioning the input space into groups, the method effectively calibrates $p(y \mid x, G)$. However, $p(y \mid x) \neq p(y \mid x, G)$, in general. Rather $p(y \mid x) = \sum_G p(y \mid x, G) p(G \mid x)$.


**Questions:**

Please respond to the concerns raised in Weaknesses above.

**Limitations:**

Yes.

---

> ### Author Rebuttal · Authors · 2023-08-08
>
>
> We express our gratitude to the Reviewer mMe6 for conducting a careful evaluation of our paper and offering valuable suggestions concerning the representation of formulas. We agree with the reviewers' perspectives, acknowledging that in certain equations, we have overloaded symbols without providing intuitive explanations, which may lead to possible misunderstandings.
>
> **Q1**: For Lemma 1 to be correct we need the additional assumption.
>
> **A1**: *We agree that although this condition is implicitly implied in our statements, a formal definition will be more intuitive and rigorous*. In Definition 1, we defined that "A grouping function $g$ is a mapping from an input $x$ to a group indicator $g(x) = G \in \mathcal{G}$". This definition implicitly requires that the function $g(x)$ is defined on any valid $x$, which is equivalent to $\forall x \in \mathcal{D},\exists G \in \mathcal{G}, s.t.  g(x)=G$. We also stated this definition explicitly in our proof of Lemma 1 in the Appendix (Line 11: "The grouping function is defined on all $x \in \mathcal{D}$"). We agree that providing an explicit formulation is more rigorous and intuitive. And we will modify this definition accordingly.
>
> **Q2**: In Definition 2 (Lines 84--90) there appears to be inconsistencies in the definition of a group G.
>
> **A2**: *Thanks for your advice and sorry for the confusion on the definition of $g(x)=G$, $x\in G$ and $(x, y)\in G$*. We use $x \in G$ and $g(x)=G$ interchangeably, that is, $x \in G \iff g(x)=G$. And the group of a labeled data $(x, y)$ is defined by the group of $x$, that is, $(x, y) \in G$ where $G=g(x)$. We clarify the definition of these symbols in our revised paper.
>
> **Q3**: Equation 13, makes no sense unless $z$ is the identity mapping (or $g$ is constant).
>
> **A3**: *We aim to introduce our implementation of $g_{\phi}(x)$ in Equation (13)*. The equation $g_{\phi}(x)=g_{\phi}(z(x))$ is meant to indicate that in our implementation, the grouping function depends on the deep feature $z(x)$ rather than raw $x$, which is a more restricted function family. Then we introduce the parameterization of $g_{\phi}(z(x))=\operatorname{softmax}(z(x)\phi_{w} +\phi_b)$. We agree that $g_{\phi}(x)=g_{\phi}(z(x))$ is an inaccurate expression and failed to reflect the difference between $g_{\phi}(x)$ and $g_{\phi}(z(x))$. We will modify Equation (13) to $g_{\phi}(x)=g^\prime_{\phi}(z(x))=\operatorname{softmax}(z(x)\phi_{w} +\phi_b)$ to highlight the difference between $g_{\phi}(\cdot)$ and $g^\prime_{\phi}(\cdot)$.
>
> **Q4**: Concerned about the correctness of the theoretical idea that underpins the approach...
>
> **A4**: *$p(y|x)=\sum_{G}p(y|x, G) p(G|x)$ is exactly how we calculate the calibrated probabilities*. As pointed out by Reviewer mMe6, by partitioning the input space into groups, the method effectively calibrates $p(y|x,G)$, which corresponds to $\hat{Y}\_{ui}$ in Algorithm 1: Line 7. Here $u$ denotes the $u$-th partition and $i$ denotes the $i$-th group in partition $u$. $\hat{Y}\_{ui}$ denotes the set of group-wisely calibrated probabilities within the group $G\_{ui}$. Then, all the group-wisely calibrated results are gathered in Algorithm 1: Line 9 to obtain calibrated results $\hat{Y}\_u$ with the $u$-th partition. All the calibrated $\hat{Y}\_u$ are averaged to obtain the final prediction $\hat{Y}\_{test}=\frac{1}{U}\sum\_u \hat{Y}\_u$. This is effectively a Monte Carlo estimation of $p(y|x)=\sum\_{G}p(y|x,G) p(G|x)$.
>
> In the revised version of the paper, we will optimize the expression of the mentioned formulas as per the reviewers' guidance and incorporate appropriate clarifications. Furthermore, we will thoroughly scrutinize all other equations in the paper to ensure the rigor and comprehensibility of all theoretical definitions and results.

---

> > ### Comment · Reviewer_mMe6 · 2023-08-17
> >
> > Thank you for your response. I have more confidence in the correctness of the approach now and will raise my score accordingly.

---

> > > ### Author Response · Authors · 2023-08-18
> > >
> > > We extend our appreciation to the Reviewer mMe6 for your thorough review of the paper and the valuable suggestions for revisions. We will proceed to enhance the paper based on your recommendations.

---

### Official Review · Reviewer_TM5v · 2023-07-02

**Soundness:** 1 poor
**Presentation:** 2 fair
**Contribution:** 2 fair
**Rating:** 5
**Confidence:** 5

**Summary:**

This work focuses on improving calibration (e.g., measured with Expected Calibration Error) using partitions of the feature space. This contrasts with most well-established recalibration techniques (isotonic regression, histogram binning, temperature scaling...) that solely use the estimated probabilities of the classifier. Partitions are learned during training with an extra linear layer + softmax on the features extracted by the deep model. Each part of the partition is recalibrated by the accuracy-preserving existing calibration techniques such as temperature scaling or ensembling temperature scaling. This ensures that the proposed method is also accuracy-preserving. The method is benchmarked on 3 datasets (CIFAR10, CIFAR100, ImageNet) and 3 networks on each dataset. The method is compared to 9 existing calibration techniques, such as temperature scaling, isotonic regression, and histogram binning.

**Strengths:**

* Learning the partitions using an extra linear layer + softmax is valuable since it is differentiable and enables end-to-end learning jointly with the network. This contrasts with existing works that use hard partitions by thresholding quantiles on a proximity metric [4-5] or by using decision trees [2-3].

**Weaknesses:**

* Positioning in literature: It misses related works that share strong conceptual links with the proposed work. Those works could have been compared either in related work or in experiments.
  * It is not compared to multicalibration [1], which proposes an algorithm for learning a multicalibrated predictor with respect to any subpopulation class.
  * The idea of partitioning the network feature space to find local miscalibration has been used before [3-4]. [3] shows that strong local miscalibration arises in modern neural networks and proposes an estimator based on feature space partitions to evaluate them. [4] links this local miscalibration to atypicality.
  * [5] also links local miscalibration to proximity and proposes an algorithm to recalibrate those subgroups. NB: [5] was released after the NeurIPS submission deadline. I put it for information.
* Framing:
  * The valuable part of finding local miscalibration and correcting them, as it is done in the proposed work, is not to improve calibration but rather to improve estimated individual posterior probabilities (i.e., reducing the epistemic loss) as pointed out in [3], or improving fairness metrics as done in [1]. The problem is that the proposed work is entirely focused on improving calibration, which is blind to local miscalibration.
  * In addition to the missing related work, the discrepancy between the current framing and the potential of the proposed method can be felt on several levels. In the introduction: "A perfectly calibrated model should be calibrated across any data space partition" (L61). This is false since a perfectly calibrated model in the standard definition of calibration (e.g., in [6] eq. (1)) just needs to be calibrated on level sets of the same predicted confidence (i.e., satisfy $\mathbb{P}(Y=\hat{Y}|\hat{\mathbb{P}} = p) = p$). However, a perfect probabilistic classifier, that is, a classifier that outputs the true individual posterior probabilities $\mathbb{P}(Y=\hat{Y}|X)$, should be calibrated on any data space partition. The exposition of the results focuses on improving ECE (Table 1 & 2).
* The improvement in calibration is marginal.
  * For example, in Table 1, the proposed method GC+TS improves on Temperature Scaling (TS) by absolute differences ranging from 1e-4 to 1e-3 in Expected Calibration Error (ECE), which is an extremely small scale for ECE. Similarly, the proposed method GC+ETS improves on Ensembling Temperature Scaling (ETS) by absolute differences ranging from 1e-4 to 1e-3.


In summary, the proposed method has the potential to improve individual posterior probabilities and fairness metrics. Unfortunately, this potential is not exploited in the current framing since it focuses on improving calibration instead. On improving calibration, the proposed method has a too marginal effect.

## References

[1] Hebert-Johnson, U., Kim, M., Reingold, O., & Rothblum, G. (2018). Multicalibration: Calibration for the (Computationally-Identifiable) Masses. In Proceedings of the 35th International Conference on Machine Learning (pp. 1939–1948). PMLR.

[2] David Durfee, Aman Gupta, & Kinjal Basu. (2022). Heterogeneous Calibration: A post-hoc model-agnostic framework for improved generalization.

[3] Alexandre Perez-Lebel, Marine Le Morvan, & Gaël Varoquaux. (2023). Beyond calibration: estimating the grouping loss of modern neural networks. ICLR.
(First released on 8 Oct 2022).

[4] Mert Yuksekgonul, Linjun Zhang, James Zou, & Carlos Guestrin. (2023). Beyond Confidence: Reliable Models Should Also Consider Atypicality.
(First released on 04 Mar 2023, https://openreview.net/forum?id=nPOKJCCvlLF)

[5] Miao Xiong, Ailin Deng, Pang Wei Koh, Jiaying Wu, Shen Li, Jianqing Xu, & Bryan Hooi. (2023). Proximity-Informed Calibration for Deep Neural Networks.
(First released on 7 Jun 2023).

[6] Chuan Guo, Geoff Pleiss, Yu Sun, & Kilian Q. Weinberger. (2017). On Calibration of Modern Neural Networks.

**Questions:**

* Could you confirm that ECE is the metric plotted in Tables 1 & 2? It is not indicated.
* Could you develop on the statistical test used for assessing the significance of the results? What are the p-values obtained with the test? Do you correct for multiple comparisons?

**Limitations:**

The authors highlight the following limitations:
* The method is restricted to deep networks since it works in the extracted feature space. It thus cannot be applied to tree models, for example.
* Increasing the number of partitions and using more complex grouping models increased the computational complexity of the method.

---

> ### Author Rebuttal · Authors · 2023-08-08
>
> I appreciate your provision of recent relevant research. Below, we will expound on the distinctions between our approach and these methods. Moreover, we intend to incorporate an analysis of these references in the revised version of the paper.
>
> **Q1**: [1] also proposes an algorithm for learning a multicalibrated predictor with respect to any subpopulation class.
>
> **A1**:
>
> 1. Although [1] proposed an algorithm to calibrate subpopulation classes, it assumes that the subpopulations are given. While our work proposes a practical method to generate useful partitions in a machine-learning manner.
>
> 2. [1] also suggests calibrating on "every subpopulation that can be identified", however, it did not reveal the underlying relationship between calibration and accuracy with the definition in [1] (which is different from ours). Since the one-to-one grouping function is a special case of "any grouping function", our example Example 4 revealed that calibrating on any grouping function is equivalent to the perfect prediction that produces groud-truth $p(y|x)$, which is generally impossible with finite data. Thus, a practical method to find a good grouping function that can improve calibration accuracy is critical, which is our core contribution.
>
> 3. [1] did not conduct any practical experiments, while the effectiveness of our method is validated in many real-world datasets and deep models.
>
> **Q2**: Comparison with [2].
>
> **A2**:  [2] proposed to group the feature space with decision trees, then apply Platt scaling within each group.  1. [2] concentrates on calibration ranking metrics (AUC), while our work concentrates on PCE metrics (ECE) and proper scoring rules (Log-Likelihood). 2. We revealed the connection between calibration and accuracy as discussed in A1.2, which is lacking in [2].
>
> **Q3**: Comparison with [3].
>
> **A3**: [3] estabilish a explained component as a metric that lower-bounding the grouping loss in the proper scoring rule theory, and conducted many experiments to validate their metric. While our work concentrates on improving the calibration and accuracy metrics. (ECE and NLL).
>
> **Q4**: Comparison with [4-5].
>
> **A4**: In the view of our framework, both [4] and [5] propose a heuristic grouping function, and apply a specific calibration method within each group. While we agree that manuually designed grouping functions may be more interpretable and may lead to better performance in some specific cases (long-tail data), we aim to propose a learning-based method that can generate partitions in an end-to-end manner.
>
> **Q5**: The problem is that the proposed work is entirely focused on improving calibration, which is blind to local miscalibration.
>
> **A5**: Sorry for the confusion. As pointed out in our paper, PCE connects calibration and accuray with grouping functions. Since the grouping functions learned by our method is neither solely probability-based (measures calibration) nor one-to-one (measures accuracy), we expected to see improvements in both view. So we report the (top-label) ECE in our paper, and NLL in the appendix Tabble 5 because of the page limit. Our method achieved statisticlly significant improvements in both ECE and NLL. We are not quite sure what is "lcoal miscalibration" mentioned by Reviewer TM5V. Experimentally, our method groups similar data, howerver, the PCE does not require such "local" grouping as discussed in response to A6 to Reviewer KvC9.
>
> **Q6**: In the introduction: "A perfectly calibrated model should be calibrated across any data space partition" (L61). This is false since a perfectly calibrated model in the standard definition of calibration (e.g., in [6] eq. (1)) just needs to be calibrated on level sets of the same predicted confidence (i.e., satisfy). However, a perfect probabilistic classifier, that is, a classifier that outputs the true individual posterior probabilities, should be calibrated on any data space partition.
>
> **A6**: [6] eq. (1) does not involve any data space parition so the definition in [6] "just needs to be calibrated on level sets of the same predicted confidence". However, when we define calibration with data space paritions, it will become different. What if each group contains $x$ with exactly the same value? The expect $y$ within each group will be exactly $p(y|x)$, which is the core idea of our Example 4. If we measure ECE within such a group, there should be only one level-set with predicted probability $q(y|x)$, and the ECE will be minimized iff. $q(y|x)=p(y|x)$.
>
> **Q7**: Could you confirm that ECE is the metric plotted in Tables 1 & 2? It is not indicated.
>
> **A7**: We report top-label ECE.
>
> **Q8**: Could you develop on the statistical test used for assessing the significance of the results? What are the p-values obtained with the test? Do you correct for multiple comparisons?
>
> **A8**: We conducted pairred t-test with 100 runs on different data splits. As reported in the paper (Table 1), following common practice, the differences with $p<0.01$ is considered to be significant.
>
>
> [1] Hebert-Johnson, U., Kim, M., Reingold, O., & Rothblum, G. (2018). Multicalibration: Calibration for the (Computationally-Identifiable) Masses. In Proceedings of the 35th International Conference on Machine Learning (pp. 1939–1948). PMLR.
>
> [2] David Durfee, Aman Gupta, & Kinjal Basu. (2022). Heterogeneous Calibration: A post-hoc model-agnostic framework for improved generalization.
>
> [3] Alexandre Perez-Lebel, Marine Le Morvan, & Gaël Varoquaux. (2023). Beyond calibration: estimating the grouping loss of modern neural networks. ICLR. (First released on 8 Oct 2022).
>
> [4] Mert Yuksekgonul, Linjun Zhang, James Zou, & Carlos Guestrin. (2023). Beyond Confidence: Reliable Models Should Also Consider Atypicality. (First released on 04 Mar 2023,
>
> [5] Miao Xiong, Ailin Deng, Pang Wei Koh, Jiaying Wu, Shen Li, Jianqing Xu, & Bryan Hooi. (2023). Proximity-Informed Calibration for Deep Neural Networks. (First released on 7 Jun 2023).

---

> > ### Comment · Reviewer_TM5v · 2023-08-19
> >
> > I thank the authors for their time in answering my comments.
> >
> > I raised my score because I think that learning the partitions in an end-to-end manner is a valuable idea worth sharing with the community and might be used in other related works. However, I am still skeptical about the contribution since the improvements, yet significant according to the tests performed, are marginals. Is this small improvement worth adding this layer of complexity? This questions the usability of the method. Also, focusing on improving ECE/accuracy quite misses the point of working with group calibration.

---

> > > ### Author Response · Authors · 2023-08-19
> > >
> > > We extend our gratitude to the Reviewer TM5v for offering numerous valuable suggestions.
> > >
> > > ### Our method is worth employing when accuracy preservation is imperative
> > >
> > > 1. Currently, there are few methods that can enhance calibration performance while maintaining accuracy. Temperature Scaling (TS) and Ensemble Temperature Scaling (ETS) persist as the current state-of-the-art methods, to a certain extent underscoring the challenge of further improvements. In this work, we introduce a pragmatic framework to optimize partitioning, yielding statistically significant improvements. This highlights the viability of this direction for enhancement. Concurrently, we acknowledge that the method we propose for learning partitioning criteria in this paper is far from optimal. *The framework we introduce could serve as a novel paradigm for refining calibration methods in the future, propelling the advancement of calibration techniques*.
> > >
> > > 2. We comprehend the reviewer's concerns regarding method complexity and performance enhancement. In our supplementary material, we conducted experimental analyses, demonstrating that our approach performs better in terms of training speed when compared with several existing methods (such as DirODIR BBQ, Beta calibration, etc.) on substantial datasets like ImageNet. In terms of implementation, we will make our source code open-source for researchers to utilize our method.
> > > Given the scarcity of accuracy-preserving techniques, *we hold the belief that our approach is worth considering for enhancing calibration performance when there is a demand to maintain accuracy*. In many cases, adding an appropriate level of complexity for marginal performance gains is indeed worthwhile.
> > >
> > > ### We do not target any specific form of local calibration
> > >
> > > We concur that employing certain local calibration metrics could potentially enhance the demonstration of our method's performance. Similar to practices in some research studies, *if we artificially define a partitioning method, we can directly optimize the calibration error within that partition, consequently enhancing the calibration performance on the corresponding partition in an intuitive manner*. In Section 3.2, "Quantitative Analysis," of the article, we indeed confirm that our method reduces the respective PCE.
> > >
> > > Nonetheless, our focus within this paper lies on more general partitioning criteria, rather than the performance on a specific partitioning standard. *The grouping function derived from our proposed method does not directly employ any artificial partitioning criteria. Consequently, its partitioning outcomes are challenging to correlate with any intuitive calibration error, such as local calibration*. However, our experiments regarding class-wise ECE, as presented in the rebuttal PDF document and section A6 addressed to Reviewer yhZK, demonstrate that our method still exhibits certain performance improvements in terms of class-level local calibration.
> > >
> > > On the other hand, *we perceive local calibration itself as a form of prior assumption, asserting that the model should be calibrated within a specific local range*. We believe that if this assumption holds true, not only would there be noticeable enhancements in our directly optimized metric (PCE), but there could also be corresponding improvements in other metrics. In essence, we introduce some prior knowledge to assist the model in predicting probabilities more accurately. Building upon this notion, we have chosen Expected Calibration Error (ECE) and Negative Log-Likelihood (NLL) as the primary evaluation metrics. The experiments indeed indicate that our approach yields significant effects on these metrics.

---

### Official Review · Reviewer_yhZK · 2023-07-06

**Soundness:** 3 good
**Presentation:** 2 fair
**Contribution:** 3 good
**Rating:** 5
**Confidence:** 5

**Summary:**

This paper proposes a method to calibrate neural networks more effectively. It introduces a general framework called PCE, which aims to provide an explanation for existing calibration methods. The proposed method partitions the input space into groups and minimizes the discrepancy between the predicted probability and the true value within each group, on average. Experimental results demonstrate that this method outperforms standard approaches like temperature scaling (TS).

**Strengths:**

- PCE, the framework introduced, offers a unified formulation for existing calibration methods. This may allow for a comprehensive understanding of these methods.
- The effectiveness of the proposed method is supported by experimental results.


**Weaknesses:**

The paper is not easy to read. I believe there is much room for improving readability.
- The term ETS is used without definition (l171)
-	It took some time to understand that "the number of partitions" is distinct from the number of groups. The study uses ‘partitions’ to mean an ensemble of multiple models. This is confusing and could be improved.
-	I could not understand how (11) is derived. How is S() determined?
-	It is difficult to interpret what g_\phi(x) learns. Suppose K=2 as in the experiments. It appears that g_\phi(x) learns binary classification using the pen-ultimate layer feature, which is also employed for the original classification task. However, the purpose and underlying explanation of this process are not clearly explained.
-	Based on my understanding, g_\phi(x) learns to partition the input space into groups that are the most uncalibrated by minimizing equation (11). Subsequently, performing temperature scaling within each of these groups and aggregating the predictions over different partitions results in improved calibration.
-	It is crucial to provide a similar, high-level explanation of how the method achieves better calibration, irrespective of whether above understanding is true or not.
-	There is no explanation what Table 1 etc. report. I assume they are top-level ECE^1 defined in [23].

Another issue is that the improvements the proposed method brings are not substantial, despite the technical complexity involved.

**Questions:**

- How can we interpret the mechanism of the proposed method that achieves a better calibration? See also Weaknesses.
- Is it sufficient to evaluate only ECE? What about class-wise ECE?
- Table 1 shows that for Swin, the original model (uncalibrated) achieves better calibration than TS/GC+TS. Calibration leads to a worse calibration. Why?


**Limitations:**

There is no serious issue regarding the limitations of this study.

---

> ### Author Rebuttal · Authors · 2023-08-08
>
>
> I express gratitude for the insightful suggestions you've provided. We will improve the paper based on your suggestions.
>
> **Q1**: The term ETS is used without definition (l171)
>
> **A1**: We will add the full name of ETS(Ensemble Temperature Scaling) and corresponding reference.
>
> **Q2**: It took some time to understand that "the number of partitions" is distinct from the number of groups.
>
> **A2**: Sorry for the confusion. Generally, a partition corresponds to a method that splits the data space into disjoint sets, while groups refer to the sets produced by a partition. We will add explanations in the revised paper accordingly.
>
> **Q3**: I could not understand how equation (11) is derived. How is S() determined?
>
> **A3**: Sorry for the confusion caused by Equation (11). *Equation (11) describes the loss used by the temperature scaling method within each group, which is not a direct estimation of PCE*. Specifically, if we choose $S$ to the average function, so the empirical estimation of $S(G_i)=\frac{1}{|G_i|}\sum_{x, y \in G_i} y$, and the empirical estimation of $S(f_{G_i}(G_i))=\frac{1}{|G_i|} \sum_{x, y \in G_i} f_{G_i}(x)$. Then, we choose the difference measure $\mathcal{L}$ to be the log-likelihood (cross entropy) $\mathcal{L}(S(G\_i), S(f\_{G\_i}(G\_i)))=\sum\_{j} S(G\_i)\_j \log S(f\_{G\_i}(G\_i))\_j$, where $j$ is the class index. The Equation (11) will be minimized by $f_{G_i}(x)=y$, which will also minimize $\mathcal{L}(S(G_i), S(f_{G_i}(G_i)))$. Thus, Equation (11) is a stronger constraint compared with minimizing PCE ($\mathcal{L}(S(G_i), S(f_{G_i}(G_i)))$) directly. Our choice of this objective is motivated by two reasons: First, Equation (11) is able to provide more signals during training since each label $y$ can guide corresponding $f_{G_i}(x)$ directly. On the contrary, if we optimize $\mathcal{L}(S(G_i), S(f_{G_i}(G_i)))$ directly, the labels and predictions are mixed and much label information is lost. Secondly, optimizing Equation (11) aligns well with the calibration method to be used. As we analyzed in A4 to Reviewer yhZK, an objective that aligns with the calibration method may lead to better calibration performance.
> In the revised paper, we shall provide a comprehensive explanation for Equation (11) to enhance clarity regarding this distinction.
>
> **Q4**: It is crucial to provide a high-level explanation of how the method achieves better calibration.
>
> **A4**: Thanks for your valuable advice. Since different groups are calibrated with different parameters, the grouping functions should find the partitions that can improve the calibrated performance with the calibration method applied within each group. We agree with Reviewer yhZK that "partition the input space into groups that are the most uncalibrated" is an intuitive and reasonable interpretation. We think a more comprehensive interpretation is that $g_{\phi}(x)$ should learn to generate partitions that best suit the calibrating methods to be used. For example, if a group has only a few elements, this group may be significantly miscalibrated (with a high ECE because of lacking of data). However, the calibration parameters (e.g., $\tau$) are prone to overfitting on this small group and result in significant miscalibration when testing. By joint training of the grouping function $g_{\phi}$ and the calibration method (temperature scaling) on the validation dataset, the grouping function is optimized to improve the overall calibration metrics (PCE), which is less likely to generate extreme partitions that may hurt
>  generalization.
>
> **Q5**: There is no explanation what Table 1 etc. report.
>
> **A5**: We report top-label ECE.
>
> **Q6**: What about class-wise ECE?
>
> **A6**: Thanks for your advice. We report class-wise ECE in the rebuttal PDF file. We can observe significant improvements on CIFAR10 and CIFAR100, while the differences on Imagenet are small. We think the reason for the performance on Imagenet is because the number of classes (1000) is significantly large than the number of partitions and groups that we have used in experiments (20 partitions and 2 groups). We have to mention that we achieve such improvements on class-wise ECE without optimizing the class-wise ECE explicitly, which shows the potential of improving calibration metrics within unknown partitions.
>
> **Q7**: Table 1 shows that for Swin, the original model (uncalibrated) achieves better calibration than TS/GC+TS. Calibration leads to a worse calibration. Why?
>
> **A7**: This is a good question that may reveal the source of performance improvements of ETS compared with TS. We visualize and compare the top-label confidence calibration of TS and ETS in Figure 1 in the rebuttal PDF. We can observe that Resnet18 tends to be overconfident and Swin tends to be underconfidence, and the top-label ECE of Resnet18 and Swin are similar. When we use TS to calibrate Resnet18, the optimal $\tau\approx 1.07$ in Resnet18 is larger than 1, which reduces overconfidence. When we apply TS to calibrate Swin, the optimal $\tau \approx 0.89$ is small than 1, which results in overconfidence in calibrated results. We may expect that the performance on Swin should improve since underconfidence is reduced. However, overconfidence is generally more dangerous for calibration metrics. For example, a predictor that always predicts a uniform distribution among all the classes will be calibrated, which corresponds to $\tau=\infty$. Generally, using a larger $\tau >1$ will push the prediction towards uniform prediction and will be more likely to reduce calibration error (regardless of overfitting). On the contrary, a $\tau<1$ that is smaller than the ground-truth $\tau^*$ may harm calibration metrics significantly which makes it even worse than uncalibrated. The main reason that ETS works much better on Swin is that ETS introduced a uniform component that can enlarge the (effective) value of$\tau$, which can reduce the potential of underestimating $\tau$ significantly.

---

### Official Review · Reviewer_BrmR · 2023-07-23

**Soundness:** 3 good
**Presentation:** 3 good
**Contribution:** 3 good
**Rating:** 7
**Confidence:** 4

**Summary:**

In this paper, the authors address the model calibration problem and propose a generalized definition of calibration error called Partitioned Calibration Error (PCE). Previous calibration methods mainly bin the data by the prediction probabilities, while PCE utilizes groups and partition functions to partition the data and requires the model to be calibrated on all partitions. The proposed calibration method jointly learns the partition function and scaling parameters to obtain the optimal calibration error. Experiment results show that the proposed method achieves significant performance improvements across multiple datasets and network architectures.

**Strengths:**

1, The proposed Partition Calibration Error provides a generalized framework for calibration error evaluation. It bridges the gap between overall accuracy and point accuracy by data partitioning. Besides, previous evaluation methods like ECE/class-wise ECE are also included in PCE, which proves that PCE is a powerful framework.
2, Based on PCE, the proposed partition calibration method is reasonable and effective. Benefiting from the generalization ability of PCE, group calibration can be applied to other calibration methods easily. The experiment results also show that group calibration can improve the performance of temperature scaling.
3, The presentation of the paper is well organized. The PCE and PC+TS are clearly introduced and important ablations are provided.

**Weaknesses:**

1, For the grouping function, it is parameterized as a set of weights and biases, which is claimed as semantic-aware. However, there might be a lot of ways to partition a set of data, such as deep feature-based clustering. The author may consider providing the reason why they choose to model the partition function as in equation (13).
2, In the experiment part, the paper only provides the results of partition calibration + (ensemble) temperature scaling. It would be helpful to test more calibration methods with partition calibration, which will demonstrate the generalization ability of partition calibration.
3, In the ablation part, the explanation of Figure 3 (a) is not sufficient. When the group number increases, the NLL and ECE also increase. The authors ascribe it to overfitting, but it might be alleviated by the proposed regularization. It would be beneficial to provide a more reasonable and insightful explanation for this result.

**Questions:**

See the weaknesses part.

**Limitations:**

The method may be limited by the computational complexity when the partition or group number is large.

---

> ### Author Rebuttal · Authors · 2023-08-08
>
> I express gratitude for the valuable suggestions you have offered to elevate the quality of our manuscript. We shall duly incorporate the corresponding modifications into the article.
>
> **Q1**: For the grouping function, it is parameterized as a set of weights and biases, which is claimed as semantic-aware. However, there might be a lot of ways to partition a set of data, such as deep feature-based clustering. The author may consider providing the reason why they choose to model the partition function as in equation (13).
>
> **A1**:   Initially, we attempted to employ k-means as the partitioning function generation approach; however, its performance showed a decline compared to TS and ETS. We attribute this observation to two potential reasons. Firstly, clustering methods inherently optimize unsupervised objectives, resulting in partitions that may not enhance performance in terms of classification and calibration metrics. In contrast, our method learns the partitioning function by utilizing log-likelihood and optimizing with validation set labels, leading to partitions directly beneficial for calibration. Secondly, during experimentation, we noticed that although k-means can generate different clustering results by setting random initial cluster centers, the final outcomes often exhibit high similarity, diminishing the performance gain from increasing the number of partitions (ensemble learning necessitates diverse predictive results). Our approach, however, introduces superior diversity.
>
> **Q2**: In the experiment part, the paper only provides the results of partition calibration + (ensemble) temperature scaling. It would be helpful to test more calibration methods with partition calibration, which will demonstrate the generalization ability of partition calibration.
>
> **A2**: The primary strength of our approach lies in enhancing the performance of accuracy-preserving calibration methods. We also conducted experiments with IRM (non-accuracy-preserving) and observed that GC+IRM did not yield performance improvements compared to IRM alone. We posit that the diminished performance of GC on IRM is primarily attributed to IRM's necessity to learn a monotonic function across all samples. With grouping, IRM can only ensure monotonicity within each group, relinquishing this property across distinct groups, which can substantially undermine its performance. Conversely, TS and ETS do not hinge on the monotonicity of predicted values, allowing them to attain better performance after grouping.
> Additionally, other methods were not accuracy-preserving and did not demonstrate any advantages in calibration metrics compared to TS and ETS, thus we did not explore them further. We acknowledge that performance improvements in the non-accuracy-preserving methods are indeed meaningful, and we intend to explore this avenue in future research.
>
> **Q3**: In the ablation part, the explanation of Figure 3 (a) is not sufficient. When the group number increases, the NLL and ECE also increase. The authors ascribe it to overfitting, but it might be alleviated by the proposed regularization. It would be beneficial to provide a more reasonable and insightful explanation for this result.
>
> **A3**: Apologies for any confusion. The concept of overfitting we mentioned here differs slightly from the general sense. In typical problems, overfitting occurs when the model's capacity is too strong, and regularization can mitigate such issues. However, in our specific problem, increasing the number of groups leads to a decrease in the number of samples within each group. As each group utilizes different model parameters, it also effectively enhances the model's capacity. While employing stronger regularization might weaken the model's capacity, we experimentally found that it is insufficient to counterbalance the negative impact caused by reduced data, resulting in models on each group tending to overfit the data within their respective groups and consequently reducing the overall generalization performance.

---

### Official Review · Reviewer_KvC9 · 2023-07-26

**Soundness:** 2 fair
**Presentation:** 3 good
**Contribution:** 2 fair
**Rating:** 6
**Confidence:** 3

**Summary:**

This paper proposes a generic form of a calibration error metric. Partitioned Calibration Error (PCE) evaluates calibration errors across the group of partitions to alleviate data uncertainty. Furthermore, the paper presents experimental results on CIFAR-10, CIFAR-100, and ImageNet using several backbone models. Interestingly, the learned partitions make visually similar groups.

**Strengths:**

- Paper is easy to follow.
- The authors propose a unified framework to evaluate confidence calibration error.



**Weaknesses:**

- The confidence calibration performance is on par with ETS.
- It would be more informative if the authors presented effects of poor partition functions for confidence calibration.
- Only vision applications are considered in this paper.
- The meaning of semantically relevance is somewhat vague.

**Questions:**

- As the learned partitions produce visually similar groups, are the features in their space proximity to each other?
- Will the performance be negatively affected if the partition function creates groups without any visual similarity?
- Does selection of hold-out data affect the performance?

**Limitations:**

Yes.

---

> ### Author Rebuttal · Authors · 2023-08-08
>
> I extend my gratitude for the queries and suggestions you've raised. We will duly enhance the relevant sections in the revised version of the article.
>
> **Q1**: The confidence calibration performance is on par with ETS.
>
> **A1**: *Our method has achieved statistically significant performance improvements in both TS and ETS*. As we primarily focus on accuracy-preserving calibration and have limited data for calibration, achieving perfect generalization calibration performance becomes challenging. With the development of calibration methods, the room for improvement in the ECE metric becomes increasingly limited, making further enhancements more difficult. In this context, our method has demonstrated statistically significant performance (with $p < 0.01$) improvements compared to baseline methods on multiple different datasets and models. Our approach not only enhances the ECE metric but also improves proper score rules, such as log-likelihood (Table 2 in rebuttal PDF), indicating that it not only improves model calibration (epistemic uncertainty)  performance but also accuracy performance concerning aleatoric uncertainty. Additionally, our method significantly improves calibration performance without compromising accuracy.
>
> **Q2**: It would be more informative if the authors presented effects of poor partition functions for confidence calibration.
>
> **A2**: This is an excellent question. *In our paper, we discuss the roles of two extreme partitioning functions and connect them using PCE*. Specifically, if we group all samples into one group, the model will predict the prior distribution for all samples, resulting in a high classification error rate but nearly perfect calibration error. On the other hand, if we use a one-to-one mapping as the partitioning function, each sample will be mapped to a different group, leading to our optimization objective degenerating into a standard log-likelihood, which is equivalent to direct fine-tuning and resulting in significant miscalibration.
>
> Efficient calibration methods proposed in prior research lie between these two extreme cases, allowing for improvements in calibration metrics. For general partitioning functions, it is challenging to intuitively evaluate which one is better. For instance, we have also attempted to use sample features for clustering (see A1 to Reviewer BrmR), but our experimental results indicate that using k-means clustering as the partitioning function leads to a decrease in calibration performance (compared to TS and ETS). This suggests that the clusters generated by the clustering may not serve as effective partitioning functions.
>
>
> **Q3**: Only vision applications are considered in this paper.
>
> **A3**: *We conducted experiments on multiple image datasets and network architectures*. We selected CIFAR10, CIFAR100, and Imagenet as the datasets, along with various network structures to cover different accuracy scenarios. For each combination of dataset and network structure, we performed 100 experiments and conducted significance testing. Exploring the calibration performance on other tasks and models is a future direction of our research.
>
> **Q4**: The meaning of semantically relevance is somewhat vague.
>
> **A4**: *In this article, our core idea is that using only probability information to construct the partitioning function is insufficient to capture information beyond the predefined classes*. However, if the partitioning function can be defined over $x$, we can learn it based on other information apart from class probabilities. In this context, semantic information emphasizes utilizing information from $x$ (beyond class probabilities) to construct the partitioning function. We acknowledge that this might lead to some misunderstanding, as class probabilities themselves can also be considered as a form of semantic information. Therefore, we will modify the wording of this part in the paper.
>
> **Q5**: As the learned partitions produce visually similar groups, are the features in their space proximity to each other?
>
> **A5**: Indeed, we added a linear layer after the features to predict the corresponding group. This can be seen as a linear partitioning of the sample space, where each group's sample features are closely located in the space. Due to the randomness in the samples and network initialization parameters, optimizing the objective function multiple times results in different partitions, thereby achieving ensemble diversity and enhancing calibration performance.
>
> **Q6**: Will the performance be negatively affected if the partition function creates groups without any visual similarity?
>
> **A6**:   As mentioned in A5, the partitions generated by our method are always close in the feature space. However, this does not imply that visually similar partitioning methods will necessarily improve performance. For example, clustering methods may produce visually similar partitions, but they might lead to worse calibration results. On the other hand, visually dissimilar partitions may not necessarily result in worse performance. For instance, if the model is overly optimistic in both class "a" and class "b" (where "a" and "b" are visually unrelated), grouping them together could help mitigate overall miscalibration issues. Thus, whether a partitioning function can improve calibration performance may be related to whether there is consistent overconfidence or underconfidence within each partition.
>
> **Q7**: Does selection of hold-out data affect the performance?
>
> **A7**: Indeed, the hold-out (HO) data significantly influences calibration performance. To ensure that the performance differences between different methods are not caused by the dataset, we randomly split the HO data and test data 100 times for each dataset-model pair. This approach allows us to conduct paired t-tests to evaluate the significance of the performance differences.

---

### Author Rebuttal · Authors · 2023-08-09

The support information of A6 and A7 to Reviewer yhZK (performance on class-wise ECE, and the reason of performance drop of TS in SWIN model), and performance measured by NLL are in the PDF file.

---

### Decision · Program_Chairs · 2023-09-21

**Decision:**

Accept (poster)

**Comment:**

The paper proposed a more generalized definition of calibration error called Partitioned Calibration Error (PCE), revealing that the key difference among these calibration error metrics lies in how the data space is partitioned. The paper was reviewed by 5 experts in the community. The authors provided detailed rebuttal and there had been extensive discussions, where most the initial major concerns have been resolved. Afterward, the paper received relatively consistent reviews as 1X "Weak Accept", 1X"Accept" and 3X"Borderline Accept".

The reviewers are conistent on the contributions of the paper while providing very important comments in how to improve the paper. The authors are requsted to revise the paper accordingly.